# Event-Guided Consistent Video Enhancement with Modality-Adaptive Diffusion Pipeline

**Kanghao Chen**[*]
AI Thrust, HKUST(GZ)
kchen879@connect.hkust-gz.edu.cn

**Zixin Zhang**[*]
AI Thrust, HKUST(GZ)
zzhang300@connect.hkust-gz.edu.cn

**Guoqiang Liang**
Nanyang Technological University

**Lutao Jiang**
AI Thrust, HKUST(GZ)

**Zeyu Wang**
CMA Thrust, HKUST(GZ)
CSE, HKUST

**Ying-Cong Chen**[†]
AI Thrust, HKUST(GZ)
EMIA, HKUST
yingcongchen@ust.hk

## Abstract

Recent advancements in low-light video enhancement (LLVE) have increasingly leveraged both RGB and event cameras to improve video quality under challenging conditions. However, existing approaches share two key drawbacks. First, they are tuned for steady low-light scenes, so their performance drops when illumination varies. Second, they assume every sensing modality is always available, while real systems may lose or corrupt one of them. These limitations make the methods brittle in dynamic, real-world settings. In this paper, we propose **EVDiffuser**, a novel framework for consistent LLVE that integrates RGB and event data through a modality-adaptive diffusion pipeline. By harnessing the powerful priors of video diffusion models, EVDiffuser enables consistent video enhancement and generalization to diverse scenarios under varying illumination, where RGB or events may even be absent. Specifically, we first design a modality-agnostic conditioning mechanism based on a diffusion pipeline by treating the two modalities as optional conditions, which is fine-tuned using augmented and integrated datasets. Furthermore, we introduce a modality-adaptive guidance rescaling that dynamically adjusts the contribution of each modality according to sensor-specific characteristics. Additionally, we establish a benchmark that accounts for varying illumination and diverse real-world scenarios, facilitating future research on consistent event-guided LLVE. Our experiments demonstrate state-of-the-art performance across challenging scenarios (*i.e.*, varying illumination) and sensor-based settings (*e.g.*, event-only, RGB-only), highlighting the generalization of our framework.

## 1 Introduction

Video enhancement is a pivotal area of research in computer vision, encompassing tasks such as denoising [32], deblurring [18], and low-light enhancement [2]. While many methods exhibit impressive performance, the inherent limitations of the sensor still constrain the video quality and realism,

---

[*]Equal Contribution
[†]Corresponding Author

39th Conference on Neural Information Processing Systems (NeurIPS 2025).

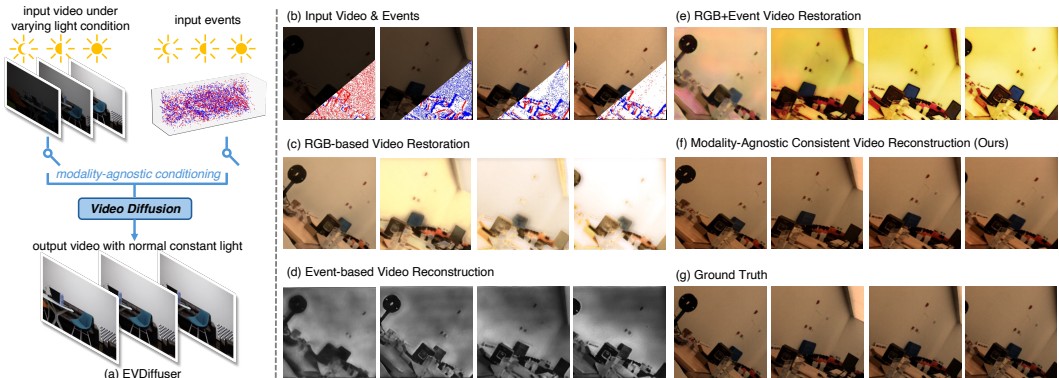

Figure 1: (a) **EVDiffuser** is a diffusion framework that enables modality-adaptive conditioning on both RGB and event data for consistent LLVE. It demonstrates robustness in real-world scenarios with varying illumination and generalizes well across multiple settings (*e.g.*, RGB-only and event-only). Under (b) challenging conditions characterized by extreme illumination variation, our (f) EVDiffuser effectively reconstructs consistent, high-quality video. In contrast, existing methods either (c) produce over-enhanced results or (d,e) fail to preserve structural details.

particularly under extreme conditions such as low light or rapid camera motion, necessitating the integration of additional sensors. Recent advancements in event cameras, which provide exceptional dynamic range and microsecond-level temporal resolution, have introduced new possibilities for video reconstruction [37, 30] and restoration [24, 23, 42]. These capabilities enable robust performance in extreme conditions, including high-speed motion, low-light environments, and overexposed scenes, thereby addressing many limitations of RGB-based approaches. Among these applications, low-light video enhancement (LLVE) has shown particular promise when leveraging event cameras.

Existing event-guided LLVE methods often assume stable input from both RGB and event modalities under steadily low illumination in a well-controlled environment. For example, recent methods [24, 23] rely on both low-light, noisy RGB frames and sufficient event data, typically constrained to a narrow illumination range (*e.g.*, 5-25 lux). However, in real-world applications, illumination can vary significantly, and different sensors are affected to varying degrees. RGB images may be of high quality most of the time, only deteriorating under specific low-light conditions. The normal quality images will still be augmented, often leading to overexposure or inconsistency in existing methods (see Fig. 1.c,e). In other cases, the RGB sensor may fail completely, capturing no usable information. The RGB+event fusion approaches may also fail to work. One stable method that can operate steadily is purely event-based event-to-video reconstruction [37, 41, 1, 9], but these methods cannot take advantage of high-quality RGB inputs when available, resulting in poor overall quality in varying illumination(see Fig. 1.d). In summary, while current techniques may perform adequately in specific settings with stable parameters (*i.e.*, illumination and modalities input), they struggle with the variability inherent in real-world conditions, where both modality availability and environmental factors frequently change. To the best of our knowledge, no existing method can effectively handle all these scenarios within a unified framework. Given these challenges, it is crucial to develop a unified, comprehensive framework that can robustly handle diverse scenarios, accurately model scene dynamics, and maintain consistent video quality despite fluctuating input conditions.

In this paper, we introduce **EVDiffuser**, a novel framework for consistent LLVE, which integrates both RGB and event data through a modality-adaptive diffusion pipeline. As shown in Fig. 1.a, the key insight of our framework is to condition the diffusion process on both modalities, producing consistent videos by leveraging the powerful priors from a pre-trained video diffusion model [51]. Specifically, we propose a modality-agnostic training pipeline based on a diffusion model, conditioning it on both modalities and fine-tuning it using augmented and integrated datasets. This facilitates the diffusion model to adapt even in the absence of either RGB or event inputs. Additionally, we design a novel modality-adaptive guidance rescaling mechanism that facilitates the integration of information from both modalities and adaptively adjusts the guidance to accommodate varying degradation conditions. This ensures the model effectively exploits the complementary characteristics of RGB and event cameras. Finally, as no existing method addresses consistent LLVE under varying illumination, we

establish a benchmark that accounts for real-world illumination variability, laying the foundation for future research across diverse applications. In our experiments, the proposed approach demonstrates strong generalization across a wide range of scenarios, maintaining consistent LLVE performance under varying illumination and showing robustness in different configurations (*i.e.*, event-only and RGB-only), even when one of the sensors is unavailable.

## 2 Related Work

### 2.1 Low-Light Video Enhancement (LLVE)

**RGB-based LLVE.** Normal-light video reconstruction with the input of low-light RGB frames has been a long-standing research topic. Although many methods [53, 4, 36, 31, 45, 10] show impressive performance, their research focuses on LLVE in *stable low-light environments*, assuming the illumination is constantly low. Some methods try to reduce temporal jitter through temporal consistency losses [4, 53, 36] and multi-frame alignment [31, 45, 10], but these are not enough for the real-world setting, where there are often unstable changes in lighting. The model design and dataset construction of current works are designed for constant low light, making these models unable to generalize to *real-world dynamic illumination*, leading to temporal discontinuities and severe distortion in the restored video.

**Multi-modal LLVE.** Recent research has explored and shown promise when combining additional sensor (*e.g.*event camera) with RGB camera to do the LLVE task. For example, Liu et al. [29] synthesize pseudo-events from adjacent low-light frames, enabling artifact-free fusion while preserving temporal consistency. EvLowLight [24] enhances coherence by jointly aligning motion and spatial features across events and frames. EvLight++ [7] further refines temporal smoothness through recurrent modules and dedicated loss functions, achieving robust noise suppression. Despite these advancements, these methods still assume a *stable low-light environment* and the *constant availability of both modalities*. They fail to consider that different sensors have varying strengths under changing lighting conditions, and they also overlook the possibility that some modalities may become unavailable in real-world scenarios, leading to unsatisfactory performance in real-world settings where illumination may fluctuate. To address this, our work aims to propose a modality-adaptive and modality-agnostic pipeline that can handle both *varying lighting conditions* and *uncertain modality availability*, respectively.

### 2.2 Video Generation Models

**Foundation Models.** The field of video generation has evolved rapidly by integrating advances in image synthesis with the challenge of ensuring temporal consistency across frames. This progress has been particularly accelerated by text-to-video models, driven by innovations in the Transformer architecture [43] and diffusion models [12]. Early works explored diverse generative frameworks: GAN-based methods [22, 34, 39] pioneered real-time video synthesis but struggled with temporal artifacts, while Transformer-VQVAE hybrids [11, 14, 50] improved long-range coherence through autoregressive token prediction, albeit at increased computational costs. Recent efforts have shifted toward diffusion-based models [51, 28, 21], which achieve state-of-the-art quality by leveraging scaled text-video datasets and architectural advancements like Diffusion Transformers (DiT) [35].

**Generative models in low-level vision.** Generative models possess strong generalization capabilities and encode rich prior information, making them widely applicable in various downstream low-level vision tasks. In the domain of multi-modal low-level vision, methods such as Event-Diffusion [26] and Temporal [58] employ diffusion models as the backbone for event-guided image reconstruction and restoration. Approaches like Repurposing [5] and EGVD [56] leverage video diffusion models combined with event cameras for frame interpolation. E2VIDiff [25] and LaSe-E2V [6] utilize prior knowledge from pretrained image diffusion models to perform event-to-video translation. In contrast to these methods, our method is built upon *pre-trained video diffusion* and introduces a modality-agnostic training pipeline by *treating the two input modalities as optional conditions*. This design enables flexible handling of different input configurations while leveraging the priors of video diffusion models to ensure consistency under dynamic lighting conditions.

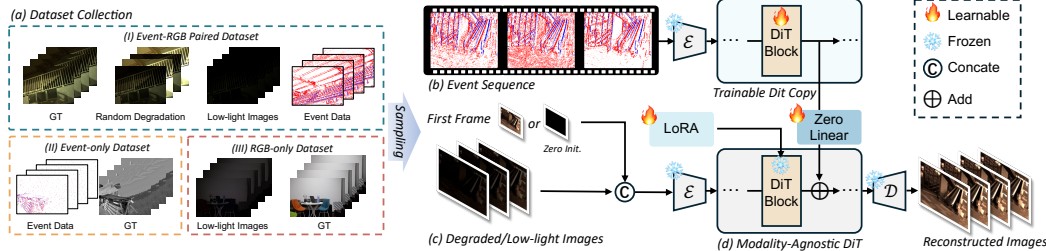

Figure 2: **Architecture of EVDiffuser**: We construct our training dataset including (I) event-RGB paired data, (II) event-to-video data, and (III) RGB-only data. To augment the dataset, random degradation is applied to the ground truth images. The (c) degraded/low-light images and (b) event sequences are processed using a (d) modality-agnostic DiT, with a trainable copy and LoRA.

## 3 Method

EVDiffuser is a diffusion framework that processes input data through a modality-adaptive pipeline, enabling seamless integration of heterogeneous modalities (*i.e.*, RGB and event data) for consistent LLVE, as illustrated in Fig. 2. Notably, our model performs robustly in dynamic scenarios with varying illumination and can operate effectively even with a single sensor when the other is unavailable. We begin by formalizing the problem statement for this novel video reconstruction task (see Sec. 3.1), followed by a review of the foundational I2V model (see Sec. 3.2). Next, we introduce our framework for injecting modality-agnostic control based on the augmented and integrated datasets (see Sec. 3.3). Finally, we present the modality-adaptive guidance rescaling technique that adaptively leverages the unique characteristic of each modality under varying conditions (see Sec. 3.4).

### 3.1 Problem Statement

We begin by defining the input data, which consists of both RGB images $\mathbf{I} \in \mathbb{R}^{H \times W \times C}$ and the corresponding event data $\mathcal{E} = \{(x^i, y^i, t^i, p^i)\}_{i=0}^{N-1}$, where $x^i$ and $y^i$ represent the pixel coordinates of each event, and $t^i$ and $p^i$ denote the timestamp and polarity, respectively. In real-world applications, the environments vary over time, such as when transitioning between indoor and outdoor scenes or between day and night. These changes can result in significant variations in illumination, thereby causing fluctuation and noise in both RGB and event data. In the worst scenario or in single-modality systems, one of the modalities may become unavailable, necessitating the modality-agnostic capability of a unified model. Given the input from both RGB and event data under varying illumination and sensor settings, the framework aims to adaptively reconstruct a consistent, high-quality video (*i.e.*, under normal-light conditions).

### 3.2 Video Diffusion Foundation Model

LLVE can be viewed as high-quality video generation conditioned on low-quality inputs. This conditional generation nature inherently aligns with video diffusion models' capability to generate high-quality video by sampling from high-dimensional video manifolds, and therefore constitutes an optimal foundation for our framework. Our EVDiffuser is fine-tuned based on the CogVideoX-I2V [51] model, a transformer-based video diffusion model [35] that operates in a latent space. Specifically, we adopt the pre-trained model as the base architecture, which takes an image $\mathbf{I} \in \mathbb{R}^{H \times W \times 3}$ as input and generates a video $V \in \mathbb{R}^{T \times H \times W \times 3}$. To prepare the input, the image $\mathbf{I}$ is first padded with zeros, resulting in a conditioned input video of size $T \times H \times W \times 3$, matching the target video's dimensions. A variational autoencoder (VAE) encoder is then applied to the input frames, producing a latent vector of size $\frac{T}{4} \times \frac{H}{8} \times \frac{W}{8} \times 16$, which is subsequently concatenated with the noise of the same size. A DiT [35] model $\epsilon_\theta$ is iteratively employed to denoise the noisy latent representation over a predefined number of steps. The resulting denoised latent vector is processed by a VAE decoder to generate the video $V$. In the following sections, we will describe how the RGB images and event data are incorporated as additional conditioning inputs to the base model, yielding our modality-agnostic model.

### 3.3 Modality-agnostic DiT for Consistent LLVE

By leveraging the strengths of the diffusion pipeline, we treat the two modalities as optional conditions and fine-tune the DiT in a modality-agnostic manner to achieve consistent LLVE. In this approach, when one modality is absent or significantly degraded, the other can serve as an optional conditioning input. Conversely, when both modalities are available, their combined information leads to a substantial improvement in video reconstruction quality. Notably, our method maintains temporal consistency despite variations in the input scenarios. The architecture overview is shown in Fig. 2(d). Below, we detail the process of injecting both modalities as conditions and fine-tuning DiT in a modality-agnostic manner.

**Injecting RGB Condition.** To incorporate the RGB sequence as a condition in the I2V diffusion model, we first apply the pre-trained VAE to obtain the corresponding latent code, which is then concatenated with the first frame latent along the frame channel. The first frame latent is introduced for auto-regressive long video reconstruction, which will be described in the following. To preserve the original DiT's powerful capabilities while adapting it to our task, we apply LoRA [15] exclusively to the image branch for parameter-efficient tuning, leaving the pre-trained DiT denoiser fully frozen.

**Injecting Event Condition.** We adopt a design similar to that of ControlNet [55] in EVDiffuser to incorporate event data as an additional conditioning input. To extract the latent features of the event sequence, we first encode the event data into a voxel grid with three channels and then apply the image VAE to encode the event data, thereby fully utilizing the pre-trained VAE. Next, we create a trainable copy of the pre-trained denoising DiT to process the latent feature of the event data. We use the first 18 blocks from the original denoising DiT to form the condition DiT for the event branch. In the event branch, we extract the output feature from each DiT block, process it with a zero-initialized linear layer, and add the resulting feature to the corresponding feature of the denoising DiT.

**Fine-tuning a Modality-agnostic DiT.** To train the DiT in a modality-agnostic manner and ensure its adaptability to varying degradation conditions and settings, we construct a comprehensive training dataset that includes videos paired with event data across multiple scenarios. The training dataset consists of three components, as shown in Fig. 2(a): **I)** First, we incorporate the RGB-event paired dataset from SDE [23] to facilitate event-guided low-light enhancement. In this setup, both low-light RGB images and event data are provided as conditioning inputs, with the model denoising them to predict the corresponding normal-light RGB sequence. To further enhance the model to handle varying illumination conditions, we introduce random degradations to normal-light RGB images, simulating dynamic lighting scenarios. These degraded images are then used as RGB inputs to reconstruct the original normal-light sequence. **II)** To enable the model to handle more extreme scenarios where the RGB input may fail entirely, we incorporate the E2V dataset from V2E2V [30], where the RGB input is replaced with zeros. **III)** Additionally, to stabilize the training of the image branch, we include a separate RGB-only low-light enhancement dataset for fine-tuning the model, where the event input is replaced with zeros accordingly. In this work, we select the SDSD dataset [45] as our RGB-only dataset. Based on these augmented and integrated datasets, the DiT is endowed with modality-agnostic capabilities and is robust to varying environments, enabling consistent LLVE. Notably, we do not use any dataset with varying illumination to train our model; nevertheless, it can seamlessly adapt to challenging, variable scenarios (see Fig. 4), demonstrating the zero-shot generalization of our framework.

Furthermore, to facilitate the reconstruction of long videos, we introduce random noise into the first frame condition and randomly replace the condition with zeros. During inference, the initial batch of the video is predicted using the zeros-initialized first frame, and the subsequent batch of video frames is predicted in an auto-regressive manner, with the final predicted frame from the previous batch serving as the first frame for the next batch.

### 3.4 Modality-adaptive Guidance Rescaling

Since our framework is based on a diffusion pipeline with modality-agnostic conditions. To further consider the unique characteristics of each modality, it is feasible to incorporate modality-adaptive guidance rescaling for each condition. To this end, Classifier-Free Guidance [13] is a promising strategy to adapt the sampling pipeline.

**Classifier-Free Guidance (CFG).** The model is trained in a modality-conditional framework, with both modalities randomly selected from the three datasets previously mentioned. Additionally, during

training, each modality is dropped with a predefined probability (*i.e.*, 0.2), facilitating the co-training of both unconditioned and conditioned models for each modality. The unconditioned model tends to be more conservative, striving to generalize across all randomly selected conditions. In contrast, the modality-conditioned model specializes in a specific environment. When applying CFG to our modality-agnostic denoising framework, we compute the discrepancy between the prediction made by the modality-specific model and the conservative model without conditioning.

**Modality-Adaptive Guidance Rescaling.** Consider a denoising model (*i.e.*, modality-agnostic DiT), denoted as $\epsilon_\theta(\mathbf{x}, \mathbf{c}_f, \mathbf{c}_e, t)$, that performs modality-adaptive generation. The CFG is implemented by adjusting the denoising process at each timestep $t$ as follows:

$$
\begin{aligned}
\hat{\epsilon}_\theta(\mathbf{x}, \mathbf{c}_f, \mathbf{c}_e, t) =& \epsilon_\theta(\mathbf{x}, \mathbf{c}_f, \varnothing, t) \\
& + \omega_e(\epsilon_\theta(\mathbf{x}, \mathbf{c}_f, \mathbf{c}_e, t) - \epsilon_\theta(\mathbf{x}, \mathbf{c}_f, \varnothing, t)) \\
& + \omega_f(\epsilon_\theta(\mathbf{x}, \mathbf{c}_f, \mathbf{c}_e, t) - \epsilon_\theta(\mathbf{x}, \varnothing, \mathbf{c}_e, t)),
\end{aligned}
\tag{1}
$$

where $\mathbf{c}_f$ and $\mathbf{c}_e$ represent the modality conditions for RGB images and events, respectively, and $\omega_*$ denotes the corresponding guidance scale. For our modality-adaptive Guidance Rescaling, $\omega_*$ is adapted to represent the degradation degree (*i.e.*, illumination) in order to exploit valuable information from each modality to accommodate the varying environment. Specifically, the illumination prior map is firstly extracted as $\mathbf{I}_{ill} = mean_{C \times H \times W}(\mathbf{I}) \in \mathbb{R}^T$, where $mean_{C \times H \times W}$ represents the operation that computes the mean values across the channel ($C$) and spatial dimensions ($H \times W$) for each frame. Then, $\omega_*$ is computed based on the illumination map, with $\omega_f = \alpha \mathbf{I}_{ill}$ and $\omega_e = \alpha(1 - \mathbf{I}_{ill})$, where $\alpha$ is a pre-define coefficient to balance the guidance. Intuitively, this formulation ensures that when illumination is low, a larger value of $\omega_e$ directs the model towards emphasizing event-to-video reconstruction, while higher illumination levels favor the frame condition.

## 4 Benchmark

We propose a benchmark tailored for practical real-world scenarios, specifically focusing on video enhancement performance under varying illumination conditions, which is largely underexplored in previous research.

### 4.1 Dataset Preparation

To construct the evaluation dataset, we synthesize videos with varying illumination based on the SDE dataset [23] and SDSD dataset [45], referred to as V-SDE and V-SDSD, respectively. Specifically, we generate frames with varying lighting conditions $V_t$ from normal-light frames $I_t$ using linear scaling: $V_t(p) = \alpha_t \times I_t(p)$, where $\alpha_t$ is a scale factor that continuously changes over time, which is set as a sine function of time. In this benchmark, we typically apply two sine wave periods in a single sequence. Moreover, to simulate noise in low-light conditions, we introduce random noise to each generated image, which is related to the its lighting level: $V_t(p) = \mathcal{N}(V_t(p), \frac{\sigma_r^2}{\alpha_t^2})$. Events are then obtained from the v2e

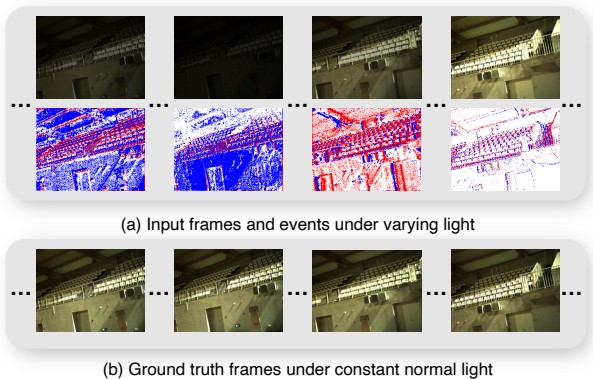

(a) Input frames and events under varying light

(b) Ground truth frames under constant normal light

Figure 3: Examples of our benchmark dataset.

model [16]. The synthesized videos and events are used as input, with normal-light videos serving as the ground truth for comparisons. An example of our test set is shown in Fig. 3.

### 4.2 Evaluation Metric

We evaluate the performance of different methods by comparing the output videos with the ground truth videos using commonly used PSNR, SSIM, and LIPIS metrics. We observe that existing

Table 1: **Comparisons on V-SDE benchmark with varying illumination.** ↑(↓) means higher (lower) is better. The highest result is highlighted in **bold** while the second highest result is in underline.

| Input | Method | V-SDE-in | | | | V-SDE-out | | | |
|---|---|---|---|---|---|---|---|---|---|
| | | PSNR↑ | PSNR*↑ | SSIM↑ | LPIPS↓ | PSNR↑ | PSNR*↑ | SSIM↑ | LPIPS↓ |
| Event Only | ETNet [47] | 14.52 | 16.13 | 0.5147 | 0.6035 | 14.21 | 16.25 | 0.4844 | 0.6217 |
| | HyperE2VID [9] | 13.77 | 16.13 | 0.4914 | 0.6398 | 13.80 | 15.94 | 0.4596 | 0.6283 |
| Image Only | SNR-Net [49] | 10.18 | 19.81 | 0.4092 | 0.4267 | 8.46 | 19.82 | 0.3830 | 0.5097 |
| | Uformer [46] | 11.04 | 19.77 | 0.5106 | 0.3672 | 8.19 | 18.88 | 0.4218 | 0.4460 |
| | LLFlow-L-SKF [48] | 11.34 | 20.04 | 0.4700 | 0.4338 | 10.50 | 20.16 | 0.4403 | 0.4331 |
| | Retinexformer [2] | 11.10 | 19.57 | 0.4330 | 0.4133 | 8.36 | 18.89 | 0.3585 | 0.5126 |
| Image+Event | ELIE [19] | 10.28 | 19.30 | 0.3835 | 0.4036 | 9.29 | 19.44 | 0.3908 | 0.4661 |
| | eSL-Net [44] | 5.81 | 20.30 | 0.3526 | 0.4232 | 5.32 | 19.26 | 0.3387 | 0.5203 |
| | Liu *et al.* [29] | 9.76 | 18.25 | 0.3519 | 0.4012 | 7.31 | 19.01 | 0.3437 | 0.4604 |
| | EvLight [23] | 10.46 | 20.17 | 0.4495 | 0.3373 | 9.06 | 19.80 | 0.4099 | 0.3914 |
| Video+Event | EvLowlight [24] | 11.29 | 20.51 | 0.4726 | 0.3698 | 9.37 | 20.33 | 0.4331 | 0.4527 |
| | EvLight++ [7] | 10.57 | 20.24 | 0.4578 | 0.3332 | 8.91 | 19.76 | 0.3995 | 0.4040 |
| | **EVDiffuser (Ours)** | **21.55** | **27.14** | **0.8338** | **0.2249** | **19.71** | **26.53** | **0.8044** | **0.1999** |

methods often overfit to low-light inputs, producing overexposed frames with normal-light inputs. To enable comprehensive comparison beyond brightness adaptation, we adopt the PSNR∗ metric from [23], which computes PSNR after aligning output and ground truth brightness.

# 5 Experiment

## 5.1 Experiment Setting

**Implementation Details.** Our model is developed by fine-tuning pre-trained I2V diffusion models (*i.e.*, CogVideoX [51]), which generate RGB videos at a resolution of 480x720 with 49 frames at 8 FPS, using 50 sampling steps. The LoRA rank is set to 128 for the image branch. For training, we use a learning rate of $1 \times 10^{-4}$ and the AdamW optimizer. The model is trained for 30 epochs with gradient accumulation, resulting in an effective batch size of 64. The training process takes 2 days on 8 H100 GPUs. During the inference stage, we employ the DDIM [40] sampler with 50 steps.

**Comparison Methods.** We compare our method with recent methods in four different settings: **(I)** the experiment with events as input, including ETNet [47] and HyperE2VID [9]. **(II)** the experiment with RGB image as input, including SNR-Net [49], Uformer [46], LLFlow-L-SKF [48], and Retinexformer [2]. **(III)** the experiment with RGB image and paired events as inputs, including ELIE [19], eSL-Net [44], Liu *et al.* [29] and EvLight [23]. **(IV)** the experiment with video and events as inputs, including EvLowLight [24] and EvLight++ [7]. Since some of the comparison methods do not release their training code, we have reproduced the training pipeline following the hyperparameters as illustrated in the reports.

## 5.2 Evaluation Results

**Comparison on V-SDE:** To evaluate performance on our V-SDE benchmark, we retrain the baseline methods (excluding event-only methods) on the SDE training dataset. The quantitative results in Tab. 1 demonstrate the superior performance of our method on the benchmark, significantly outperforming the comparison methods across all metrics. For the original PSNR, existing methods show a sharp decline due to their tendency to overfit to the illumination conditions in the training set, resulting in poor generalization to diverse real-world lighting. To assess image restoration effectiveness excluding the effect of illumination fitting, we compare PSNR∗, where our method again significantly outperforms SOTA techniques, achieving improvements of 6.63 dB for V-SDE-in and 6.20 dB for V-SDE-out. For SSIM and LPIPS, our method also achieves superior performance, providing strong evidence for its effectiveness in low-light image enhancement. Notably, the event-only method shows relatively higher PSNR among the baselines, as it relies solely on event data to generate videos, making it more robust to illumination variation. Additionally, we provide an evaluation based on metrics from VBench [17] in the Appendix (see Tab. 7), to further demonstrate the effectiveness of our method on temporal consistency. There is also a notable observation that many methods using both event and image modalities perform worse than single-modality approaches on PSNR. For event-only methods, since these methods input only event data, they are not sensitive to image

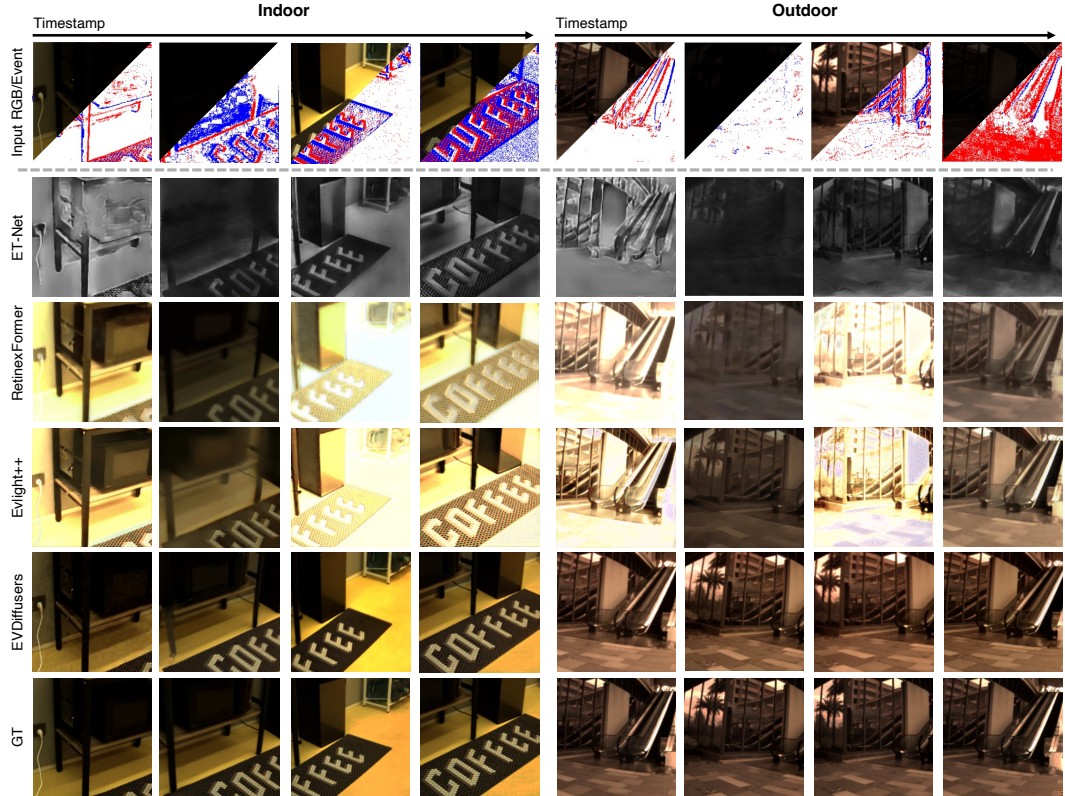

Figure 4: Qualitative results on the V-SDE benchmark under varying illumination.

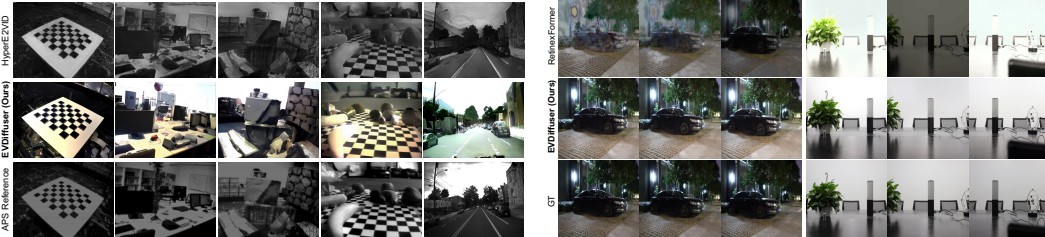

Figure 5: Qualitative comparison with state-of-the-art methods on event-to-video dataset, including ECD [33] (Column 1&2&3), HQF [41] (Column 4) and MVSEC [57] (Column 4).

Figure 6: Qualitative comparison with state-of-the-art methods on RGB-only dataset (*i.e.*, SDSD [45]).

intensity variations across different scenarios. The intensity of reconstructed frames remains stable within a certain range, resulting in relatively higher PSNR compared to image-involved methods, which are severely affected by varying frame intensities. Additionally, we compare PSNR* with intensity calibration to evaluate the structural fidelity of reconstructed videos, where image-involved methods achieve better performance than event-only methods. When comparing image-only and multi-modality methods, we observe that EvLowlight and EvLight++ achieve better performance, while other methods show comparable performance. This is because these methods typically overfit to specific low-light conditions and mostly fail to adapt to varying illumination scenarios.

Qualitatively, as shown in Fig. 4 for both indoor and outdoor scenes, our method consistently reconstructs sequences under varying illumination, preserving clear edges and producing high-quality results that closely resemble the ground truth. In contrast, competing methods either suffer from overexposure under normal lighting or fail to capture details when RGB data is heavily degraded.

Table 2: **Comparisons on our V-SDSD dataset with RGB-only setting**. The highest result is highlighted in **bold** while the second highest result is highlighted in underline.

| Method | V-SDSD-in | | | V-SDSD-out | | |
|---|---|---|---|---|---|---|
| | PSNR*↑ | SSIM↑ | LPIPS↓ | PSNR*↑ | SSIM↑ | LPIPS↓ |
| SNR-Net [49] | 18.96 | 0.6056 | 0.4155 | 19.52 | 0.4543 | 0.4656 |
| Uformer [46] | 18.69 | 0.6894 | 0.2997 | 19.07 | 0.5153 | 0.3555 |
| LLFlow-L-SKF [48] | 20.46 | 0.6485 | 0.5070 | 20.98 | 0.4780 | 0.6404 |
| Retinexformer [2] | 19.83 | 0.6738 | 0.2525 | 19.68 | 0.4622 | 0.3509 |
| **EVDiffuser (Ours)** | **29.10** | **0.9163** | **0.1033** | **25.23** | **0.6365** | **0.1927** |

We also evaluate our method on the original SDE test set, as shown in the Appendix (see Tab. 8), to further demonstrate its generalization capability.

To assess the effectiveness of our work in real-world scenarios, we collect a sequence with varying illumination across indoor and outdoor scenes. While it is impractical to capture high-quality RGB for these scenes, we provide only qualitative results in the Appendix (see Fig. 8).

**Comparison on Event-only Setting:** Since the dynamic range of RGB sensors is relatively lower than that of event sensors (*e.g.*, 60 dB vs. 120 dB), the RGB sensor may fail entirely under extremely low illumination. To demonstrate the generalization capability of our method in extreme cases and its modality-agnostic capability, we conducted comparisons on event-to-video reconstruction datasets, including ECD [33], MVSEC [57], and HQF [41]. We did not retrain the model on additional datasets, thus evaluating its zero-shot capability. In these scenarios, existing RGB+event methods fail without RGB input due to their reliance on both modalities. Therefore, we compare our method only with HyperE2VID [9], a SoTA E2V method, as shown in Fig. 5. Although our method may introduce additional textures in under-constrained regions lacking sufficient event data, it is able to recover high-quality details and appearance, producing a more realistic scene compared to HyperE2VID.

**Comparison on RGB-only setting:** It is worth noting that our modality-agnostic framework also functions without event input, enhancing its practicality for deployment in systems without event sensors. To further evaluate the modality-agnostic capability of our framework, we assess its performance on an RGB-only benchmark (*i.e.*, V-SDSD), reflecting scenarios where no event camera is available on the system. To compare, we retrain the baseline models using the SDSD training dataset. As shown in Tab. 2 and Fig. 6, our method outperforms the baselines in terms of PSNR*, SSIM, and LPIPS, achieving gains of over 8.64 dB for V-SDSD-in and 4.25 dB for V-SDSD-out.

Table 3: Impact of each training dataset.

| Event+RGB | RGB only | Event only | PSNR↑ | SSIM↑ | LPIPS↓ |
|---|---|---|---|---|---|
| ✓ | ✗ | ✗ | 18.31 | 0.7797 | 0.2542 |
| ✓ | ✓ | ✗ | 18.51 | 0.7839 | 0.2555 |
| ✓ | ✓ | ✓ | 21.55 | 0.8338 | 0.2249 |

Table 4: Impact of the LoRA adapter.

| PEFT Strategies | PSNR↑ | PSNR*↑ | SSIM↑ | LPIPS↓ |
|---|---|---|---|---|
| Frozen | 21.17 | 25.86 | 0.7936 | 0.2533 |
| w LoRA | 21.55 | 27.14 | 0.8338 | 0.2249 |

## 5.3 Ablation Study

**Impact of the Integrated Datasets.** To assess the impact of the fusion in our training dataset, we conduct experiments with multiple dataset combinations. As shown in Tab. 3, training exclusively on the Event+RGB dataset yields a PSNR of 18.31 dB, whereas incorporating the RGB-only dataset results in an improvement of 0.2 dB. Including all three types of datasets achieves the best results, demonstrating that the diffusion model benefits from a diverse and comprehensive dataset to enhance performance for consistent LLVE.

**Impact of the LoRA for Image branch.** To verify this, we conduct an ablation study in Tab. 4. Compared to the baseline with a frozen image branch, incorporating LoRA leads to a 0.38 dB increase in PSNR, demonstrating the importance of the adapter. This is because the original model is pre-trained on the image-to-video task, while our model focuses on video-to-video. We also experiment with fully fine-tuning all parameters; however, this approach encounters memory limitations.

**Impact of the Modality-aware Guidance Rescaling.** In this section, we conduct an ablation study to assess the impact of each individual modality and evaluate the effectiveness of our proposed Modality-aware Guidance Rescaling (MGR). As shown in Fig. 7, the event-only baseline is able to recover the scene's edges but fails to capture the full visual appearance, often leading to hallucinated details. Similarly, the image-only baseline produces moderate results in normal conditions, although it may underperform in extreme scenarios. Additionally, the model incorporating both modalities without MSR exhibits degraded visual quality, as it overly relies on the degraded images. In contrast, when MSR is applied, the model shows significant improvement. This comparison highlights the effectiveness of our MGR and emphasizes the flexibility of the diffusion-based framework in managing event-based applications.

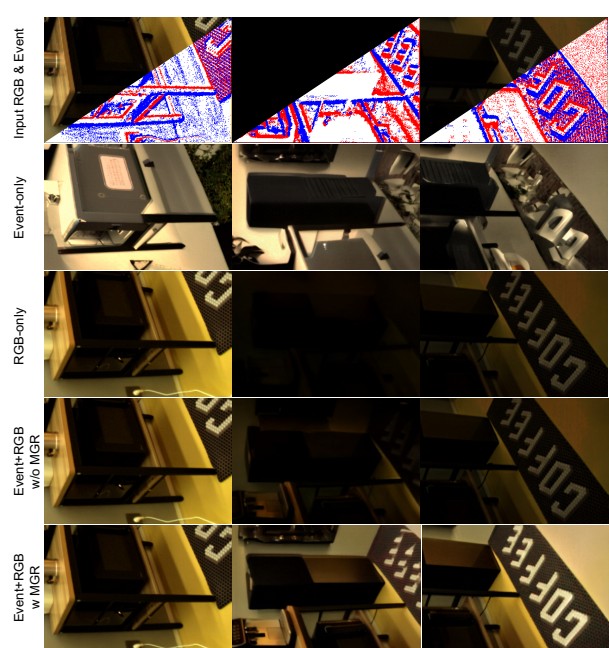

Figure 7: Impact of modality inputs and our Modality-adaptive Guidance Rescaling (MGR).

## 6    Conclusion

We present a novel framework for consistent LLVE that seamlessly integrates both RGB and event data through a modality-agnostic diffusion pipeline. Our approach guarantees high-quality reconstruction with robust temporal consistency, even in dynamic and challenging environments. Experimental results demonstrate the superiority of our framework, offering a solid foundation for future advancements in addressing real-world variability.

**Limitations and Future Work:** Due to the inherent limitations of the diffusion model, our inference process requires multiple sampling steps, which can be time-consuming. This affects the efficiency of our framework in applications with strict performance requirements. In future work, we aim to explore more efficient diffusion models for this framework, including one-step sampling and quantization techniques, to reduce computational costs. Additionally, while our framework is specifically tailored for the LLVE task — one of the promising applications of event cameras — it can potentially be adapted for other tasks (*e.g.*, interpolation, denoising) by leveraging the strengths of our approach.

**Broader Impact:** The proposed EVDiffuser framework has significant potential for real-world impact. By enabling consistent, high-quality video enhancement under challenging conditions—including low light, dynamic illumination, and partial sensor failure—it can benefit applications in surveillance, autonomous driving, and wearable devices. Its modality-agnostic design enhances resilience and flexibility, promoting broader deployment even when only a single sensor is available. This adaptability fosters accessibility in resource-constrained settings and paves the way for more reliable vision systems across diverse environments, ultimately contributing to safety, security, and situational awareness in both public and private sectors.

## Acknowledgements

This work is supported by National Natural Science Foundation of China (No. 62206068) and Guangdong Provincial Key Lab of Integrated Communication, Sensing and Computation for Ubiquitous Internet of Things (No.2023B1212010007). This work was partially supported by the Guangdong Provincial Key Lab of Integrated Communication, Sensing and Computation for Ubiquitous Internet of Things #2023B1212010007 and the Guangzhou Industrial Information and Intelligent Key Laboratory #2024A03J0628.

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

# Appendix

## A User Study

We conducted a user study with 16 participants evaluating the performance of different methods. We employ preference ranking to assess user satisfaction across different methods. The evaluation focuses on four key aspects: lighting continuity, video quality, detail preservation, and similarity to the ground truth (GT similarity). As shown in Tab. 5, across all subjects and in every aspect, our method stood out with a consistently superior preference score, highlighting that it is perceived by humans as significantly superior to current baseline methods.

Table 5: User Study Preference Ranking

| Method | Lighting Continuity ↓ | Video Quality ↓ | Detail Preservation ↓ | GT Similarity ↓ |
|---|---|---|---|---|
| ET-Net [47] | 4.375 | 4.938 | 5.000 | 4.875 |
| Retinexformer [2] | 3.813 | 3.563 | 3.063 | 3.688 |
| Evlight++ [7] | 3.063 | 2.438 | 2.688 | 2.813 |
| EvLowlight [24] | 2.688 | 3.063 | 3.125 | 2.563 |
| **EVDiffuser (Ours)** | **1.000** | **1.000** | **1.125** | **1.063** |

## B Runtime Profile and Parameter Count

Tab. 6 summarizes the efficiency profile of our method. While our unified design entails a higher parameter count and computational cost compared to modality-specific models, it delivers **significantly improved generalization** across a wide range of challenging scenarios. We prioritize generalization and performance over efficiency, and our method focuses on **offline use** as a post-processing technique, rather than for real-time applications.

Moreover, our proposed framework is inherently **extensible and compatible** with emerging acceleration techniques for diffusion-based generative models. Below, we highlight several promising directions:

- **Quantized Attention:** Recent advancements [54] in quantized attention mechanisms have demonstrated a $1.8\times$ speedup for CogVideoX without degradation in video quality. These techniques are fully compatible with our CogVideoX-based architecture.

- **Flow Matching:** The recent flow matching algorithm [20] in video generation reinterprets the denoising process as a multi-stage pyramidal flow, achieving real-time video generation at 24 FPS for 768p resolution. This approach can be seamlessly adapted to our diffusion model with a modality-agnostic training pipeline.

- **Auto-regressive Modeling:** Recent studies [3, 27] have explored auto-regressive strategies for video generation, formulating the task as next-token prediction and achieving real-time sampling at 24 FPS. Extending these techniques to event camera presents a promising direction for fully leveraging the high temporal resolution inherent to event data.

- **One-step Diffusion:** Recent efforts [8, 52, 38] have shown progress in reducing the number of denoising steps for generation tasks (achieving a 12 FPS), although they remain limited in handling complex video-generation scenarios. In our preliminary exploration, we implemented a naive few-step diffusion pipeline within our framework, reducing the sampling process to 20 DDIM steps. However, this reduction introduces a trade-off between computational efficiency and video quality, which we recognize as a valuable direction for future work.

In summary, while our unified framework incurs additional computational cost, it opens up a meaningful research topic and provides a flexible and generalizable solution across modalities and illumination conditions. Furthermore, the aforementioned acceleration techniques are immediately applicable to our proposed EvDiffuser framework and constitute a central focus of our future work. We hope our response helps to clarify the core contributions of our work and resolves the concern.

Table 6: Efficiency profile and parameter count of different methods.

| Method | Params (M) | Memory (GB) | Inference Time (s/frame) |
|---|---|---|---|
| LLFlow-L-SKF [48] | 409.50 | 39.91 | 0.30 |
| Retinexformer [2] | 15.57 | 1.61 | 0.12 |
| ELIE [19] | 440.32 | 33.36 | 0.32 |
| eSL-Net [44] | 560.94 | 0.56 | 0.18 |
| EvLight++ [7] | 225.91 | 26.21 | 0.21 |
| **EVDiffuser (Ours)** | 7932.80 | 30.89 | 5.63 |

## C    Additional Implementation Details

For the comparison method, we reproduced ELIE [19], Liu *et al.* [29] and Evlight++ [7] according to the implementation details in the original papers, while the others are retrained with the released code. We replace the event synthesis module in [29] by inputting events captured with the event camera or generated from the event simulator [16].

## D    Qualitative Results on the Self-capture Sequence

To further demonstrate the effectiveness of our method in real-world scenarios, we record a sequence under varying illumination conditions, covering both outdoor and indoor environments. As shown in Fig. 8, our method consistently preserves clear structure and appearance across frames, including outdoor scenes with normal lighting and indoor scenes where the RGB input nearly fails. In contrast, the comparison methods perform well only under specific illumination conditions and fail in most frames, underscoring the limitations of existing approaches.

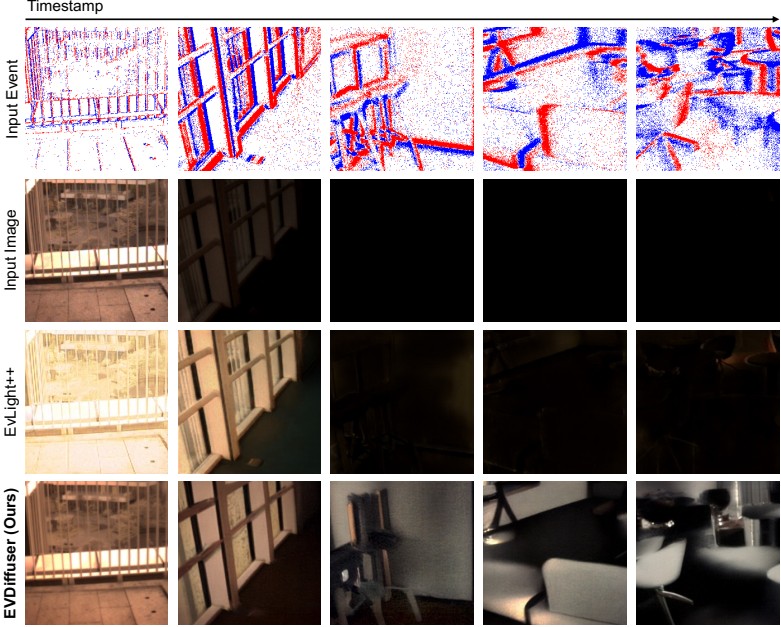

Figure 8: Qualitative results on self-captured sequence under varying illumination, when moving from outdoor to indoor scenes.

## E    Quantitative Comparison on Temporal Consistency

We conduct a comprehensive evaluation of video enhancement quality by assessing the temporal consistency metric introduced in VBench [17]. As shown in Tab. 7, our method achieves significant

improvements in capturing temporal coherence, demonstrating the effectiveness of our modality-agnostic EVDiffuser for consistent LLVE.

Table 7: **Comparisons on the temporal metric from VBench [17].** A higher score indicates relatively better performance for a particular dimension.

| Methods | Subject Consistency | Background Consistency | Motion Smoothness |
|---|---|---|---|
| ELIE [19] | 0.6597 | 0.8716 | 0.9841 |
| eSL-Net [44] | 0.6382 | 0.8992 | 0.9842 |
| Liu *et al.* [29] | 0.6468 | 0.8784 | 0.9815 |
| EvLowlight [24] | 0.6183 | 0.9040 | 0.9824 |
| EvLight++ [23] | 0.6610 | 0.8793 | 0.9881 |
| EVDiffuser (Ours) | **0.7007** | **0.8823** | **0.9942** |

## F    Additional Qualitative Results on V-SDE Dataset

Fig. 9 shows a comprehensive qualitative comparison between our method and baseline methods, featuring the proposed V-SDE Dataset. Our method significantly preserves both image consistency and quality, producing results that closely match the ground truth acquired under constant lighting conditions.

## G    Comparison on the Original SDE and SDSD Datasets

We also test our modality-agnostic model on the original SDE and SDSD datasets, where stable low light condition is maintained across the sequences. As shown in Tab. 8, although our model is not specifically designed for stable low light condition, our method still achieves performance comparable to the SoTA under this setting, demonstrating the generality of our approach. Notably, the comparison methods are specifically designed for this stable low-light condition, limited to handling only the illumination conditions consistent with their training sets.

Table 8: **Comparisons on the original SDE dataset [23]**. The highest result is highlighted in **bold** while the second highest result is highlighted in underline.

| Method | SDE-in | | | SDE-out | | |
|---|---|---|---|---|---|---|
| | PSNR↑ | PSNR*↑ | SSIM↑ | PSNR↑ | PSNR*↑ | SSIM↑ |
| SNR-Net [49] | 20.05 | 21.89 | 0.6302 | 22.18 | 22.93 | 0.6611 |
| Uformer [46] | 21.09 | 22.75 | 0.7524 | 22.32 | 23.57 | 0.7469 |
| LLFlow-L-SKF [48] | 20.92 | 22.22 | 0.6610 | 21.68 | 23.41 | 0.6467 |
| Retinexformer [2] | 21.30 | 23.78 | 0.6920 | 22.92 | 23.71 | 0.6834 |
| ELIE [19] | 19.98 | 21.44 | 0.6168 | 20.69 | 23.12 | 0.6533 |
| eSL-Net [44] | 21.25 | 23.19 | 0.7277 | 22.42 | 24.39 | 0.7187 |
| Liu *et al.* [29] | 21.79 | 23.88 | 0.7051 | 22.35 | 23.89 | 0.6895 |
| EvLight [23] | 22.44 | 24.81 | 0.7697 | 23.21 | 25.60 | 0.7505 |
| EvLowlight [24] | 20.57 | 22.14 | 0.6217 | 22.04 | 23.72 | 0.6485 |
| EvLight++ [7] | 22.67 | 25.83 | 0.7791 | 23.34 | 26.01 | 0.7676 |
| **EVDiffuer (ours)** | 20.03 | 22.34 | 0.6537 | 22.76 | 24.32 | 0.7298 |

## H    Impact of Variation Frequencies

We also conduct an analysis study to analyze the impact of varying illumination change frequencies in video sequences. Here, we define frequency as the number of sine wave periods occurring within a single sequence. As shown in Tab. 9, as the change frequency increases, the performance metrics PSNR and PSNR* decrease accordingly, while other metrics fluctuate within a narrow range. This demonstrates the robustness of our method to changes in illumination frequency, as our training process incorporates more extreme and randomized augmentations.

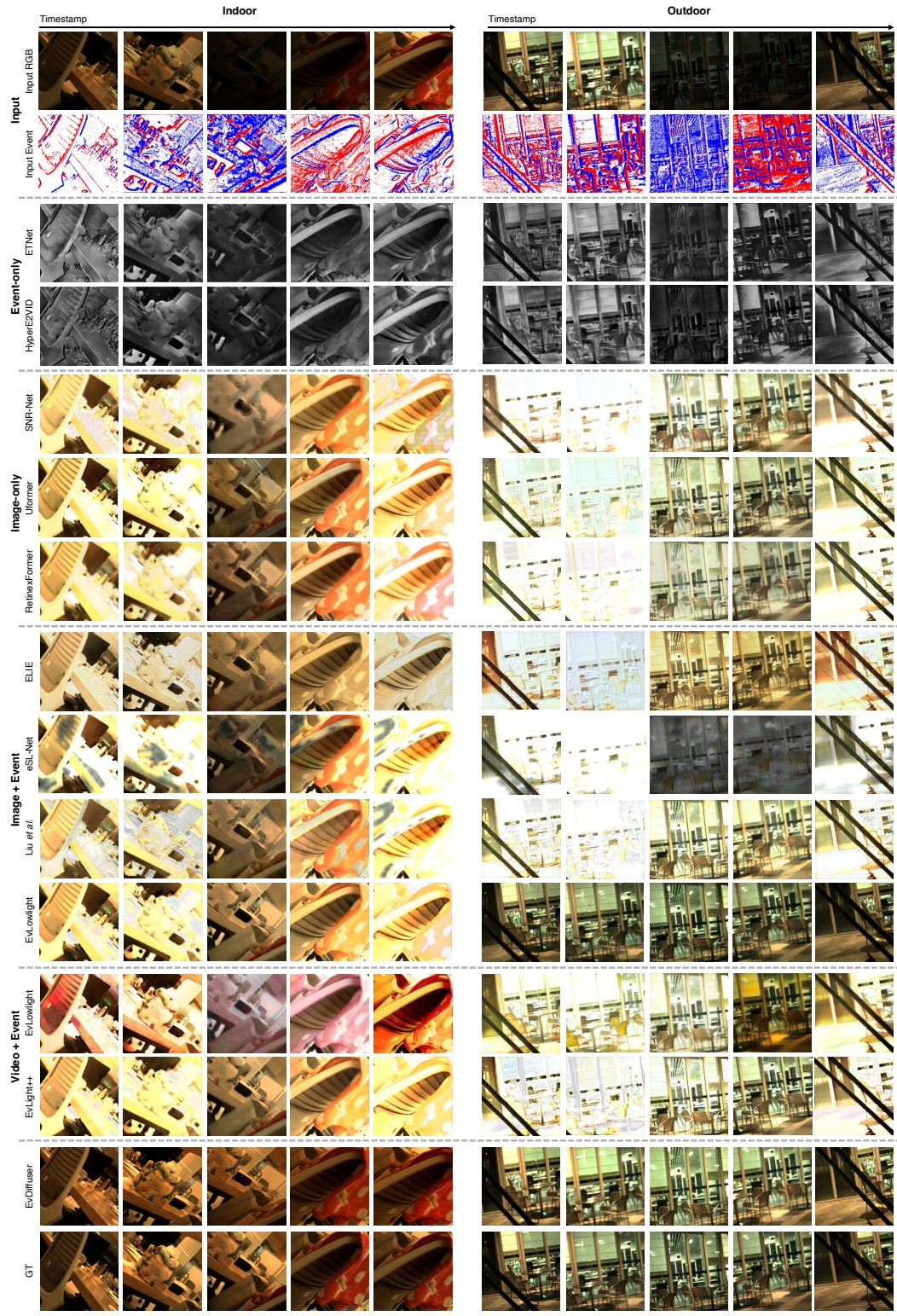

Figure 9: Additional qualitative comparison on V-SDE Dataset.

Table 9: **Analysis study on the changing frequency of illumination.** Frequency is defined as the number of sine wave periods occurring within a single sequence.

| Frequencies | V-SDE-in | | | |
|:---:|:---:|:---:|:---:|:---:|
| | PSNR↑ | PSNR*↑ | SSIM↑ | LPIPS↓ |
| 1 | 22.89 | 27.47 | 0.8158 | 0.2148 |
| 2 | 21.55 | 27.14 | 0.8338 | 0.2249 |
| 3 | 20.76 | 26.89 | 0.8278 | 0.2279 |

# I  Impact of the Random degradation

We also perform an ablation study to assess the impact of random augmentation in illumination. As shown in Fig. 10, in the absence of random augmentation, the illumination varies in parallel with the input RGB, which highlights the effectiveness of the random degradation strategy.

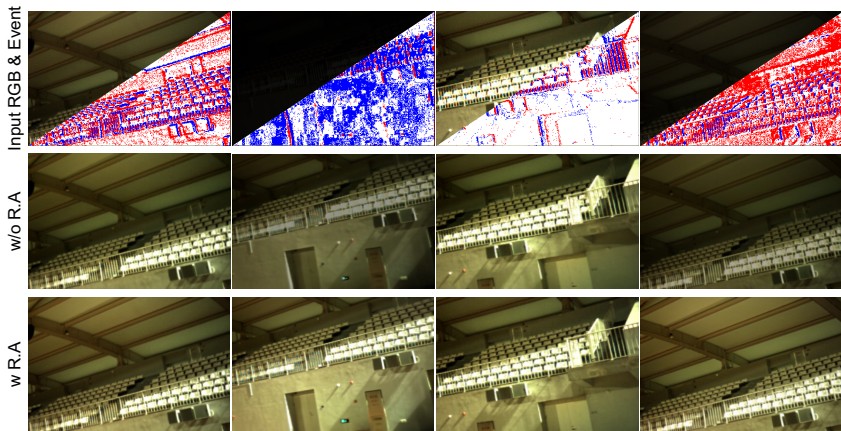

Figure 10: Impact of the random augmentation (RA) in illumination.

