# OpenReview forum: "Event-Guided Consistent Video Enhancement with Modality-Adaptive Diffusion Pipeline"
_NeurIPS.cc/2025/Conference — NeurIPS 2025 poster_

### Official Review · Reviewer_j9ey · 2025-06-20

**Clarity:** 3
**Significance:** 1
**Originality:** 2
**Rating:** 4
**Confidence:** 5

**Summary:**

This paper proposes an event-guided video enhancement method targeting varying lighting conditions. To achieve modality-agnostic reconstruction from both event and image data, two modality-agnostic DiT models are introduced—one for each modality. Additionally, to account for the distinct characteristics of each modality, a Modality-Adaptive Guidance Rescaling strategy is proposed for classifier-free guidance. The method is validated on both synthetic and real datasets to demonstrate its effectiveness. While it outperforms existing methods under varying lighting conditions, it does not surpass state-of-the-art methods on the low-light enhancement dataset SDE, as shown in Table 6.

**Questions:**

1. A comparison with existing methods trained on the same dataset should be included to ensure fair evaluation.
2. Performance comparisons in terms of runtime, FLOPs, and the number of parameters should be provided to assess computational efficiency.
3. The results for the "Event only" and "Image only" settings should be included in Table 1 to better understand the contribution of each modality.
4. The smoothness effects introduced by the proposed method should be further explained and analyzed, particularly regarding their impact on visual quality and detail preservation.

**Ethical Concerns:**

["NO or VERY MINOR ethics concerns only"]

**Final Justification:**

The authors have addressed parts of my concerns. Although I still have reservations regarding the slow runtime and occasional blurriness artifacts, I acknowledge that this work represents a valuable attempt toward advancing low-light video enhancement.

**Limitations:**

Yes

**Quality:**

3

**Strengths And Weaknesses:**

The proposed method explicitly accounts for the differences between events and images by formulating a two-branch pipeline for reconstruction. It significantly outperforms existing methods in both qualitative and quantitative evaluations under varying lighting conditions, demonstrating superior performance in consistency preservation.
However, several concerns remain:
1. Unfair Training Conditions: The proposed method is trained on a specifically designed dataset aimed at consistency preservation under varying lighting conditions, whereas the compared methods are trained under consistent lighting. This raises the question of whether the observed improvements stem from the dataset or the model itself. A fair comparison—where all methods are trained on the same dataset—is necessary to better isolate and validate the contribution of the proposed model.
2. Efficiency Concerns: The use of two DiT models raises concerns regarding computational cost. A comparison of runtime, FLOPs, and the number of parameters with existing methods should be included to assess the efficiency of the proposed approach.
3. Ablation of Input Modalities: The proposed method can operate with "Event only" and "Image only" inputs, yet the results for these settings are missing in Table 1. Including this comparison would help demonstrate the robustness and flexibility of the model when only one modality is available.
4. Detail Preservation: There are concerns regarding the model’s ability to preserve fine details. For example, in “Indoor Scene 3” from the supplementary video, sharp details appear overly smoothed by the proposed method. This issue warrants further analysis and discussion.

---

> ### Author Rebuttal · Authors · 2025-07-31
>
> We sincerely thank the reviewer for the thorough evaluation and constructive feedback. Below, we provide detailed responses to each point.
>
> **Q1: Unfair Training Conditions**
>
> Sorry for the confusion. There may be a misunderstanding regarding the training pipeline of EvDiffuser and the comparison methods. We would like to clarify the following points:
>
> 1) **Overall Framework:** First of all, we clarify that our method addresses a novel and realistic problem: **modality-adaptive** video enhancement under **various lighting conditions**. Both the model architecture and the dataset are designed **synergistically** to tackle this problem.
>
> 2) **Dataset Integration:** To enable the model to handle different modality availabilities, our dataset integrates RGB, event, and RGB-event paired data. Based on a **modality-agnostic training pipeline**, our approach can effectively learn from all these data types. In contrast, the comparison methods are modality-specific and can only utilize a subset of the dataset, making **direct training under integrated dataset infeasible**. For example, training EvLowlight (event+RGB) on an RGB-only dataset is meaningless because it relies on valid inputs from both modalities for interaction. Similarly, Retinexformer (RGB) cannot be trained on the RGB-event dataset, as it is incapable of processing event data.
>
> 3) **RGB Augmentation:** To further handle diverse illumination conditions, our model applies random degradations to RGB images during training (as illustrated in Line 179). To validate this, we performed an additional comparison by applying the same augmentation strategy to the training sets of RGB-based baseline methods. As shown in the table below, although the baselines show modest improvements, they still struggle to adapt to varying illumination because their modality-interaction mechanisms are not designed for such conditions.
>
> | Method     | PSNR $\uparrow$ | PSNR* $\uparrow$ | SSIM $\uparrow$ | LPIPS $\downarrow$ |
> |------------|------|------|------|-------|
> | Retinexformer | 16.89 | 24.10 |0.6815 | 0.2540 |
> | EvLight++  | 15.42 | 24.93 |0.6980 |0.3167 |
> | EvLowlight (Ours) |21.55 | 27.14 | 0.8338 |0.2249 |
>
> In conclusion, our research focuses on modality-adaptive video enhancement under diverse lighting conditions. The model design, training pipeline, and dataset are all developed in synergy to address this task. The additional comparison using similar RGB augmentation for baseline methods further validates the effectiveness of our approach. We hope this clarifies the misunderstanding and addresses your concerns.
>
>
>
>
> **Q2: Computational Efficiency Analysis**
>
> We clarify that our framework does not utilize two entire DiT models. Instead, building on the pre-trained CogVideoX DiT backbone (42 blocks), we employ only the first **18 blocks** as the conditioning branch, as described in Line 169. This approach significantly reduces computational overhead while preserving effectiveness.
>
> The table below summarizes the efficiency profile of our method. Compared to traditional frameworks, our diffusion-based approach requires more parameters but achieves significantly better generalization across diverse scenarios.
>
> | Method        | Params (M)  | Memory (GB) | Inference Time (s/frame) |
> |---------------|------------|-----------|-----------|
> | LLFlow-L-SKF  | 409.50     | 39.91     | 0.30
> | Retinexformer | 15.57      | 1.61      | 0.12
> | ELIE          | 440.32     | 33.36     | 0.32
> | eSL-Net       | 560.94     | 0.56      | 0.18
> | EvLight++     | 225.91     | 26.21     | 0.21
> | EvLowlight (Ours)    | 7932.80    | 30.89     | 5.63
>
>
> **Q3: Single-Modality Performance Results**
>
> Fig. 7 have shown the qualitative results of EVDiffuser under single-modality configurations.
> Quantitatively, we provide results for EVDiffuser on the V-SDE-out dataset. These results demonstrate comparable performance in both event-only and RGB-only configurations compared to existing methods, validating our framework's robustness and flexibility. We will incorporate these results into Table 1 in the revision.
>
> |Input | Method     | PSNR $\uparrow$ | PSNR* $\uparrow$ | SSIM $\uparrow$ | LPIPS $\downarrow$ |
> |------|------------|------|-------|------|-------|
> | Event-only | ETNet | 14.21 |16.25 |0.4844 |0.6217 |
> |            | HyperE2VID | 13.80 |15.94 |0.4596 |0.6283 |
> |            | **EVDiffuser (ours)** | 13.69 | 16.11 | 0.4856 | 0.5920 |
> | RGB-only   | LLFlow-L-SKF | 10.50 | 20.16 | 0.4403 | 0.4331 |
> |            | Retinexformer | 8.36 |18.89 |0.3585 |0.5126 |
> |            | **EVDiffuser (ours)** | 17.20 | 24.48 | 0.6997 | 0.2102 |
> | Event+RGB | EvLight | 9.06 | 19.80 | 0.4099 | 0.3914 |
> |           | EvLowlight | 9.37 |20.33 |0.4331 |0.4527 |
> |           | **EVDiffuser (ours)** | 19.71 |26.53| 0.8044 |0.1999 |
>
>
>
> **Q4: The smoothness effects introduced by the proposed method**
>
> We respectfully disagree with the assessment of smoothness in the *“Indoor Scene 3”* example. Our method successfully reconstructs fine details, such as the **electric wire and its shadow**, whereas existing methods exhibit significant artifacts and loss of detail. Quantitative evaluations in Table 1 further demonstrate that our approach achieves higher PSNR and SSIM scores, indicating superior detail preservation compared to existing methods.
> Additionally, we conducted a user study with 16 participants to evaluate the performance of different methods. We employed preference ranking to assess user satisfaction based on four key aspects: lighting continuity, video quality, detail preservation, and similarity to the ground truth (GT similarity).
> Across all participants and evaluation criteria, our method consistently achieved the best preference score, showing that it is perceived by users as significantly superior to current baseline methods. We will include these results in the revised manuscript.
>
>
> | Method | lighting continuity $\downarrow$ | Video Quality $\downarrow$ | Detail Preservation $\downarrow$ | GT similarity $\downarrow$ |
> | --- | --- | --- | --- | --- |
> | ET-Net | 4.375 | 4.9375 | 5 | 4.875 |
> | Retinexformer | 3.8125 | 3.5625 | 3.0625 | 3.6875 |
> | Evlight++ | 3.0625 | 2.4375 | 2.6875 | 2.8125 |
> | EvLowlight | 2.6875 | 3.0625 | 3.125 | 2.5625 |
> | EvLowlight (Ours) | **1** | **1** | **1.125** | **1.0625** |

---

> > ### Comment · Reviewer_j9ey · 2025-08-02
> >
> > I totally appreciate the authors' responses, which have addressed parts of my concerns. However, I still have the following remaining questions:
> >
> > 1. Q2: Although the performance of the proposed method is impressive, the large number of parameters and extremely slow inference speed (reportedly ~20× slower than existing methods) present significant limitations for real-world deployment. Is there any possibility to optimize the model to achieve at least real-time inference (e.g., ≥1 FPS)? I am aware of some recent advances in one-step diffusion methods that substantially reduce inference time while maintaining quality—could such techniques be adopted here?
> >
> > 2. Q4: I acknowledge the results of the user study, which show that the proposed method achieves the highest scores in terms of *lighting continuity*, *video quality*, *detail preservation*, and *GT similarity*. However, I remain concerned about the **noticeable blurriness** in some frames (e.g., at **0:35** and **1:04** in the supplementary video). These frames appear significantly blurrier than others, **despite the input frames being sufficiently sharp**. Could the authors explain the source of this inconsistency?

---

> > > ### Author Response · Authors · 2025-08-03
> > >
> > > We sincerely thank the reviewer for the further discussion.
> > >
> > > **Q2: Computational Efficiency**
> > >
> > > Our framework is primarily designed to adapt a pre-trained video diffusion model to modality-agnostic training pipeline. This design is inherently **flexible** and can readily incorporate future advances in diffusion-based generative modeling to enhance both efficiency and performance.
> > > We acknowledge the potential of one-step diffusion methods as a promising direction. However, our method builds upon CogVideoX — a widely adopted open-source video diffusion model — which currently does not support one-step sampling techniques. Moreover, to the best of our knowledge, the majority of recent one-step diffusion methods [1,2] are tailored to image-based diffusion models, often relying on model distillation or adversarial training frameworks. Extending these techniques to video generation remains non-trivial, where the added time dimension complicates the ODE trajectories and increases computational demands.
> > > Although some recent efforts [3] have shown progress in reducing the number of denoising steps for text-to-video tasks (achieving a 12 FPS), they remain constrained to low-resolution outputs and simple motion patterns. In our preliminary exploration, we implemented a naive few-step diffusion pipeline within our framework, reducing the sampling process to 20 DDIM steps. However, this reduction introduces a trade-off between computational efficiency and video quality, which we recognize as a valuable direction for future work.
> > >
> > > Beyond one-step diffusion, we also identify several promising avenues to further enhance the computational efficiency of our framework:
> > > * **Quantized Attention:** Recent advancements [4] in quantized attention mechanisms have demonstrated a 1.8× speedup for CogVideoX without degradation in video quality. These techniques are fully compatible with our CogVideoX-based architecture.
> > >
> > > * **Flow Matching:** The recent flow matching algorithm [5] in video generation reinterprets the denoising process as a multi-stage pyramidal flow, achieving real-time video generation at 24 FPS for 768p resolution. This approach can be readily adapted to our modality-agnostic training pipeline.
> > >
> > > * **Auto-regressive Modeling:** Recent studies [6,7] have explored auto-regressive strategies for video generation, formulating the task as next-token prediction and achieving real-time sampling at 24 FPS. Extending these techniques to event camera presents a promising direction for fully leveraging the high temporal resolution inherent to event data.
> > >
> > > In summary, while we agree that one-step diffusion is a valuable goal, current methods are not yet mature for complex video tasks. We believe the alternative directions outlined above are immediately promising and are fully compatible with our proposed EvDiffuser. We are excited to explore these challenging avenues in our future work.
> > >
> > >
> > > [1] One-step diffusion with distribution matching distillation
> > >
> > > [2] Adversarial diffusion distillation
> > >
> > > [3] Dollar: Few-step video generation via distillation and latent reward optimization
> > >
> > > [4] Sageattention2: Efficient attention with thorough outlier smoothing and per-thread int4 quantization
> > >
> > > [5] Pyramidal flow matching for efficient video generative modeling
> > >
> > > [6] Diffusion forcing: Next-token prediction meets full-sequence diffusion
> > >
> > > [7] Autoregressive Adversarial Post-Training for Real-Time Interactive Video Generation
> > >
> > > **Q4: Blurriness in Some Frames**
> > >
> > > We sincerely thank the reviewer for the meticulous feedback.
> > > * **Regarding 0:35**: We acknowledge the presence of a trailing artifact on the white wall to the left. This occurs because the input frame in this segment is almost entirely dark (not sharp), while the event data captures plausible but noisy structural cues. Consequently, our EvDiffuser relies heavily on the event data for reconstruction in these regions, which can lead to artifacts in smooth areas (e.g., the white wall) where event signals are sparse or absent. However, in a horizontal comparison, other methods are less effective at preserving fine details, and struggle with lighting consistency.
> > >
> > > * **Regarding 1:04**: We have carefully review the video frame-by-frame but were unable to identify noticeable blur in this segment. We would greatly appreciate it if the reviewer could provide more specific details about the region in question. It is also worth noting that the video demo was encoded at 720p to meet file size requirement, which may have introduced slight quality degradation. We will include the original high-definition video in the revision.
> > >
> > > Overall, while we acknowledge that our method is not flawless and may exhibit artifacts in extremely challenging scenarios, we believe that EvDiffuser demonstrates superior performance in detail preservation and temporal consistency compared to existing approaches. We also recognize that there is room for further improvement, and we will explore these enhancements in future work.

---

### Official Review · Reviewer_LDfn · 2025-07-03

**Clarity:** 3
**Significance:** 3
**Originality:** 3
**Rating:** 4
**Confidence:** 5

**Summary:**

This paper introduces EVDiffuser, a novel framework for consistent low-light video enhancement (LLVE) that adaptively integrates RGB and event camera data through a modality-adaptive diffusion pipeline. EVDiffuser can handle varying illumination and operate effectively even when one modality is unavailable. Comprehensive experiments demonstrate that EVDiffuser significantly outperforms state-of-the-art methods across various challenging scenarios, including varying illumination and sensor-based settings.

**Questions:**

- Why did you choose to encode events into a three-channel voxel grid and use a pre-trained image VAE without fine-tuning? Given the significant modality gap between event data and RGB images, wouldn't fine-tuning the VAE for event data be more appropriate?
- What motivated the decision to use different approaches for injecting RGB (LoRA) and event data (ControlNet-style)? Did you perform ablation studies comparing alternatives, such as using ControlNet for both modalities or LoRA for both?
- What is the maximum sequence length your model can handle in the auto-regressive long video reconstruction mode? Do you observe error accumulation over time, and if so, how do you address this issue?
- Could you clarify what you mean by "modality-agnostic"? Have you tested or do you have theoretical reasons to believe your framework would work with other modalities beyond RGB and event data, such as infrared, depth, or point cloud data?
- In Table 1, many methods using both event and image modalities perform worse than single-modality approaches. What explains this counterintuitive result?
- Given the high computational requirements of diffusion models, how does your approach compare to baselines in terms of FLOPS, runtime, parameters, and throughput? Are there any strategies you've explored to make the inference more efficient?
- Since your experiments use synthetically generated low-light data, how confident are you about the model's generalization to real-world low-light scenarios with complex noise patterns and degradations?
- How does EVDiffuser compare to powerful contemporary diffusion models like Stable Diffusion or Wan (https://huggingface.co/Wan-AI/Wan2.1-T2V-14B) in terms of video quality and generation capabilities?

**Ethical Concerns:**

["NO or VERY MINOR ethics concerns only"]

**Limitations:**

yes

**Quality:**

3

**Strengths And Weaknesses:**

Strengths:
- The paper addresses a significant gap in LLVE research by introducing a unified framework that handles varying illumination and adapts to different input modality configurations.
- The evaluation covers a wide range of scenarios (varying illumination, event-only, RGB-only) and compares against numerous state-of-the-art methods.

Weaknesses:
- In line 166, the authors mention converting event data into a three-channel format, but this approach ignores the significant modality gap between event data and RGB images. Using the pre-trained image VAE directly on this converted event data without fine-tuning is problematic and may not effectively capture the unique characteristics of event data.
- The paper uses a ControlNet-style approach for event data injection but LoRA for RGB injection without providing adequate justification or ablation experiments for this design choice. This lack of explanation makes the architectural decisions seem arbitrary rather than principled.
- While the authors mention auto-regressive long video reconstruction, they don't specify the maximum sequence length the model can handle, whether error accumulation occurs over time, or how such issues are addressed.
- The term "modality-agnostic" is somewhat misleading. It's unclear whether it means different modalities aren't important or that the model can accept various modalities as input. The paper only validates RGB and event modalities, making claims about broader applicability to infrared images, depth maps, or point clouds seem overreaching.
- The paper primarily integrates existing techniques (diffusion models, ControlNet, LoRA) without substantial innovation in the diffusion framework itself. The inclusion of Classifier-Free Guidance (CFG) as a methodological contribution is questionable since it's a well-established diffusion training technique that belongs in the experimental details rather than the method section.
- In Table 1, many methods using both event and image modalities perform worse than single-modality approaches, which seems counterintuitive and requires explanation.
- As a diffusion-based approach, the method requires multiple sampling steps during inference, making it computationally expensive. The paper lacks comprehensive comparisons of computational metrics (FLOPS, runtime, parameters, throughput) with baselines. The mention of training on 8 H100 GPUs (line 257) suggests a prohibitively high computational cost compared to baselines.
- The low-light data used in experiments is synthetically generated, raising questions about generalization to real-world low-light scenarios.
- The paper lacks comparisons with powerful contemporary diffusion models like Stable Diffusion or Wan (https://huggingface.co/Wan-AI/Wan2.1-T2V-14B), which would provide better context for the method's performance.
- I am also curious about the results in the case where both event and RGB are missing. The model is still able to generate promissing results?

---

> ### Author Rebuttal · Authors · 2025-07-31
>
> We sincerely thank the reviewer for the thorough evaluation and constructive feedback. Below, we provide detailed responses to each point.
>
> **Q1: Event data encoding and VAE usage without fine-tuning**
>
> We appreciate this concern regarding our event data processing. First, the three-channel conversion technique is widely adopted in existing methods. Regarding the potential domain gap and the use of a pre-trained VAE for encoding event frames, we evaluated the reconstruction performance and observed a **sufficiently low reconstruction error** (MSE: 0.0076), demonstrating that the pre-trained VAE can effectively encode event frames.
> Technically, the VAE is trained on large-scale image data and primarily serves as an encoder to project event frames into latent space. The unique characteristics of event data are captured and **learned within the conditional DiT branch**. Also, fine-tuning the VAE on event data remains a promising direction for future work and could further enhance performance.
>
>
>
>
>
> **Q2: Different injection approaches for RGB (LoRA) and event data (ControlNet-style)**
>
> Table 4 presents the results without LoRA for the image branch. Technically, our architectural design follows modern practices for conditioning diffusion models.
>
> * **RGB conditioning:** We condition the diffusion backbone by concatenating each input RGB frame with the noisy latent. Since the image-to-video CogVideoX is pre-trained with similar conditioning strategy to extract RGB features, it naturally facilitates extending the diffusion prior to our low-light images. Leveraging this property, we repurpose the generic video-generation model for low-light video enhancement via a lightweight LoRA adaptation, thereby retaining the original diffusion priors while introducing only a minimal parameter overhead.
>
> * **Event conditioning:** We employ a ControlNet-style conditioning approach, which is a widely adopted technique in recent conditional diffusion models. The ControlNet branch is essential for handling a completely different domain not learned by the pre-trained CogVideoX, similar to recent ControlNet applications (e.g., sketch-based and depth-based generation).
>
> These design choices enable us to effectively leverage the prior knowledge of the pre-trained video diffusion model while adapting it to the specific requirements of our modality-agnostic low-light video enhancement. We will clarify this rationale in the revised version.
>
>
>
>
> **Q3: Auto-regressive long video reconstruction limitations**
>
> We have demonstrated long-video results (~300 frames) in our supplementary video demo. Technically, unlike existing image-to-video frameworks that rely solely on the first frame and **lack intermediate control**, our approach is constrained by sequential frames and event data. To further prevent over-reliance on the initial frame, we **randomly add noise and mask to the frames** during training (Line 191), encouraging the model to leverage both event streams and RGB information more effectively.
> Overall, these design choices substantially reduce error propagation compared with conventional first-frame-based auto-regressive methods.
>
> **Q4: "Modality-agnostic" terminology clarification**
>
> We appreciate this concern. By *“modality-agnostic,”* we mean that our framework can operate effectively and adaptively under varying modality availability, including cases where certain modalities are partially corrupted or entirely absent. This differs from modality-invariant approaches, which assume equal contributions from all modalities. We will make this clearer in the final version.
> In this work, we focus exclusively on RGB and event data for consistent low-light video enhancement. For modalities beyond RGB and event data, we believe that our modality-agnostic training pipeline is flexible enough to be extended to incorporate additional modalities for diverse downstream tasks, as different modalities can be treated as different conditions to the diffusion backbone. We will discuss this in the final version.
>
>
> **Q5: Limited methodological innovation in diffusion framework**
>
> We would like to re-iterate the innovation of our framework. While we build upon modern diffusion techniques, our contributions lie in the novel integration and adaptation of existing techniques for the challenging consistent LLVE task. We facilitate a modality-adaptive diffusion framework that enables consistent LLVE across varying illumination and modality availability for the first time. Technically, we propose a modality-adaptive training pipeline that injects both modalities and fine-tunes on integrated and augmented datasets. Additionally, we present a well-designed modality-adaptive guidance rescaling mechanism that dynamically adjusts guidance based on illumination conditions and modality reliability. Overall, **the model design, training pipeline, and dataset are all developed in synergy to address this task**.
>
> **Q6: Counterintuitive multi-modal performance in Table 1**
>
> Thank you for this question. This is indeed a surprising observation that many methods using both event and image modalities perform worse than single-modality approaches.
> For event-only methods, since these methods input only event data, they are not sensitive to image intensity variations across different scenarios. The intensity of reconstructed frames remains stable within a certain range, resulting in relatively higher PSNR compared to image-involved methods, which are severely affected by varying frame intensities. Additionally, we compare PSNR* with intensity calibration to evaluate the structural fidelity of reconstructed videos, where image-involved methods achieve better performance than event-only methods.
> When comparing image-only and multi-modality methods, we observe that EvLowlight and EvLight++ achieve better performance, while other methods show comparable performance. This is because these methods typically overfit to specific low-light conditions and mostly fail to adapt to varying illumination scenarios.
>
> **Q7: Computational efficiency concerns**
>
> The table below summarizes the efficiency profile of our method. Compared to traditional frameworks, our diffusion-based approach requires more parameters but achieves significantly better generalization across diverse scenarios.
> For efficiency improvements, we have tried one-step diffusion and flow matching, which can reduce sampling to 20 DDIM steps. However, we recognize this represents a **trade-off between computational efficiency and performance**, which may be a promising future direction.
>
>
> | Method        | Params (M)  | Memory (GB) | Inference Time (s/frame) |
> |---------------|------------|-----------|-----------|
> | LLFlow-L-SKF  | 409.50     | 39.91     | 0.30
> | Retinexformer | 15.57      | 1.61      | 0.12
> | ELIE          | 440.32     | 33.36     | 0.32
> | eSL-Net       | 560.94     | 0.56      | 0.18
> | EvLight++     | 225.91     | 26.21     | 0.21
> | EVDiffuser (Ours)    | 7932.80    | 30.89     | 5.63
>
> **Q8: Generalization to real-world low-light scenarios**
>
> Sorry for the confusion. We have included real-world self-captured sequence for qualitative evaluation in the supplementary material (Fig. 8).
> When transitioning from outdoor to dark indoor environments, our method successfully reconstructs videos with consistent lighting, even under extremely low-light conditions, thanks to the modality-agnostic framework.
> In contrast, existing methods either overexpose outdoor scenes or fail to reconstruct extremely low-light regions, performing well only within a narrow illumination range. This further demonstrates our method’s strong generalization capability in realistic scenarios.
> Due to the time limit, we will collect additional real-world data in the future and include it in the revised version.
>
>
> **Q9: Comparison with contemporary diffusion models**
>
> Direct comparison is not practical because EVDiffuser is designed for a different task than general video generation models (e.g., Wan). These large-scale models serve as powerful **foundation models** and can provide stronger backbones for EVDiffuser compared to CogVideoX. We attempted to adapt the Wan model for our task, and although it achieved better video quality, it incurred an excessively high computational cost, making it impractical for us.
>
>
>
> **Q10: Performance without both modalities**
>
> The model can still generate reasonable results when both RGB and event data are missing. This is because our model is based on the pre-trained CogVideoX and fine-tuned by randomly masking both modalities for CFG inference. However, this scenario is impractical for our LLVE task.

---

### Official Review · Reviewer_18Kq · 2025-07-03

**Clarity:** 4
**Significance:** 4
**Originality:** 3
**Rating:** 4
**Confidence:** 4

**Summary:**

This paper introduces EVDiffuser for consistent low-light video enhancement (LLVE) that integrates RGB and event camera data via a modality-adaptive diffusion pipeline. It features a modality-agnostic conditioning mechanism and a modality-adaptive guidance rescaling strategy, enabling high-quality, temporally consistent video reconstruction even when RGB or event data are missing or corrupted. Extensive experiments on challenging benchmarks show that EVDiffuser significantly outperforms state-of-the-art methods, demonstrating superior generalization and resilience in real-world dynamic environments.

**Questions:**

1. Could the authors clarify the computational requirements and complexity of the proposed method, such as inference time and GPU memory usage, both during training and inference?

2. The current evaluation is conducted on synthetic dynamic illumination conditions. Are there any results available on real-captured datasets with varying lighting? I understand that conducting such experiments may be difficult within the rebuttal period, so it will not affect my decision if these results cannot be provided. However, it is important to note that the noise patterns of event cameras can differ substantially under real-world lighting variations, which might not be fully captured by v2e simulation. Since the paper primarily claims advantages in real-world scenarios, providing results on real-captured sequences with dynamic lighting would be crucial and could significantly strengthen the paper by demonstrating its generalization ability.

**Ethical Concerns:**

["NO or VERY MINOR ethics concerns only"]

**Final Justification:**

I have read the rebuttal and the discussions with other reviewers. It appears there is a shared understanding that, despite the significant computational overhead, this work introduces a meaningful new research perspective. As a result, I maintain my original positive evaluation.

**Limitations:**

yes

**Quality:**

3

**Strengths And Weaknesses:**

Strengths:

1. Addressing the realistic challenge of varying illumination and potential modality corruption is highly valuable and practical.

2. The experimental results are encouraging and demonstrate strong potential.

3. The implementation of the proposed idea is clear, well-structured, and reasonable.

Weaknesses:

1. The experimental data is purely synthetic, which may limit the persuasiveness of the results.

2. Employing two DiT models could make the overall framework computationally expensive during both training and inference.

---

> ### Author Rebuttal · Authors · 2025-07-31
>
> We sincerely thank the reviewer for the thorough evaluation and constructive feedback. Below, we provide detailed responses to each point.
>
> **Q1: The experimental data are purely synthetic, which may limit the persuasiveness of the results.**
>
> Sorry for the confusion. We have included real-world self-captured sequence for qualitative evaluation in the supplementary material (Fig. 8).
> When transitioning from outdoor to dark indoor environments, our method successfully reconstructs videos with consistent lighting, even under extremely low-light conditions, thanks to the modality-agnostic framework.
> In contrast, existing methods either overexpose outdoor scenes or fail to reconstruct extremely low-light regions, performing well only within a narrow illumination range. This further demonstrates our method’s strong generalization capability in realistic scenarios.
> Following your suggestion, we will collect additional real-world data in the future and include it in the revised version.
>
>
>
>
> **Q2: Employing two DiT models could make the overall framework computationally expensive during both training and inference.**
>
> We acknowledge this concern and would like to clarify our architectural design. Our framework does not utilize two entire DiT models. Instead, building on the pre-trained CogVideoX DiT backbone (42 blocks), we employ only the first 18 blocks as the conditioning branch, as described in Line 169. This approach significantly reduces computational overhead while preserving effectiveness.
>
>
> The computational requirements are detailed below:
>
> ***Training Requirements:***
> - GPU Memory: 76GB for batch size 4
> - Training Time: ~2 days on 8×H100 GPUs
>
> ***Inference Requirements:***
> - GPU Memory: 31GB
> - Inference Time: 5.63 seconds per frame on a single H100 GPU
>
> Additionally, the table below summarizes the efficiency profile of our method. Compared to traditional frameworks, our diffusion-based approach requires more parameters but achieves significantly better generalization across diverse scenarios.
>
> | Method        | Params (M)  | Memory (GB) | Inference Time (s/frame) |
> |---------------|------------|-----------|-----------|
> | LLFlow-L-SKF  | 409.50     | 39.91     | 0.30
> | Retinexformer | 15.57      | 1.61      | 0.12
> | ELIE          | 440.32     | 33.36     | 0.32
> | eSL-Net       | 560.94     | 0.56      | 0.18
> | EvLight++     | 225.91     | 26.21     | 0.21
> | EVDiffuser    | 7932.80    | 30.89     | 5.63

---

> > ### Comment · Reviewer_18Kq · 2025-08-04
> > **Thank You for the Rebuttal**
> >
> > I thank the authors for the detailed response. That is a considerable amount of parameters and runtime, which suggests that the proposed method is only suitable for offline use as a post-processing technique, rather than for real-time applications. In the latter case, alternative solutions might be more practical. For example, using a lightweight classifier to assess lighting conditions and applying existing low-light processing methods accordingly. For modality-agnostic handling, one could train separate models for RGB-only, event-only, and dual-modality inputs, and select the appropriate model based on input quality. While such a solution is more engineering-driven, it raises the question of whether the substantial computation increase (10× in parameters and nearly 20× in runtime) is justified solely for integrating everything into a single model.
> >
> > Still, this paper opens up a new research topic, which I believe is meaningful. I therefore maintain my original positive evaluation.

---

> > > ### Author Response · Authors · 2025-08-04
> > >
> > > We sincerely thank the reviewer for the insightful feedback and discussion.
> > >
> > > The suggestion to employ an ensemble of modality-specific models is indeed valuable, especially regarding the computational efficiencies and real-time applications. However, we argue that such a framework introduces inherent limitations that our unified model is designed to mitigate:
> > >
> > > 1. **Temporal Consistency**: An ensemble of separate models would struggle to maintain temporal consistency across dynamic environments. When switching between models under varying lighting conditions, the model lacks a shared temporal context across models, leading to potential inconsistencies (e.g., abrupt transitions between frames) under extreme illumination changes.
> > >
> > > 2. **Scalability and Data Utilization**: This approach is inherently difficult to scale-up. Each specialized model is trained on its own modality-specific data, preventing the system from leveraging on the rich, correlated information within multiple multimodal datasets.
> > > In contrast, our unified end-to-end model is flexible to be trained on an integrated dataset comprising RGB, event, and paired RGB-event data, which enables our model to effectively exploit shared multimodal knowledge.
> > >
> > > Overall, while we agree that using separate models represent an alternative solution, it faces inherent limitations that warrant further exploration and mitigation. In contrast, we consider the unified model paradigm to be a crucial and promising direction. In this way, several advanced acceleration techniques can be readily integrated into our framework to enhance computational efficiency:
> > >
> > > * **Quantized Attention:** Recent advancements [1] in quantized attention mechanisms have demonstrated a 1.8× speedup for CogVideoX without degradation in video quality. These techniques are fully compatible with our CogVideoX-based architecture.
> > >
> > > * **Flow Matching:** The recent flow matching algorithm [2] in video generation reinterprets the denoising process as a multi-stage pyramidal flow, achieving real-time video generation at 24 FPS for 768p resolution. This approach can be seamlessly adapted to our diffusion model with a modality-agnostic training pipeline.
> > >
> > > * **Auto-regressive Modeling:** Recent studies [3,4] have explored auto-regressive strategies for video generation, formulating the task as next-token prediction and achieving real-time sampling at 24 FPS. Extending these techniques to event camera presents a promising direction for fully leveraging the high temporal resolution inherent to event data.
> > >
> > > * **One-step Diffusion:** Recent efforts [5,6,7] have shown progress in reducing the number of denoising steps for generation tasks (achieving a 12 FPS), although they remain limited in handling complex video-generation scenarios. In our preliminary exploration, we implemented a naive few-step diffusion pipeline within our framework, reducing the sampling process to 20 DDIM steps. However, this reduction introduces a trade-off between computational efficiency and video quality, which we recognize as a valuable direction for future work.
> > >
> > > We believe the acceleration techniques outlined above are immediately promising and are fully compatible with our proposed EvDiffuser. This remains a key priority for our future work.
> > >
> > > [1] Zhang J, Huang H, Zhang P, et al. Sageattention2: Efficient attention with thorough outlier smoothing and per-thread int4 quantization[J]. arXiv preprint arXiv:2411.10958, 2024.
> > >
> > > [2] Jin Y, Sun Z, Li N, et al. Pyramidal flow matching for efficient video generative modeling[J]. arXiv preprint arXiv:2410.05954, 2024.
> > >
> > > [3] Chen B, Martí Monsó D, Du Y, et al. Diffusion forcing: Next-token prediction meets full-sequence diffusion[J]. Advances in Neural Information Processing Systems, 2024, 37: 24081-24125.
> > >
> > > [4] Lin S, Yang C, He H, et al. Autoregressive Adversarial Post-Training for Real-Time Interactive Video Generation[J]. arXiv preprint arXiv:2506.09350, 2025.
> > >
> > > [5] Ding Z, Jin C, Liu D, et al. Dollar: Few-step video generation via distillation and latent reward optimization[J]. arXiv preprint arXiv:2412.15689, 2024.
> > >
> > > [6] Yin T, Gharbi M, Zhang R, et al. One-step diffusion with distribution matching distillation[C]//Proceedings of the IEEE/CVF conference on computer vision and pattern recognition. 2024: 6613-6623.
> > >
> > > [7] Sauer A, Lorenz D, Blattmann A, et al. Adversarial diffusion distillation[C]//European Conference on Computer Vision. Cham: Springer Nature Switzerland, 2024: 87-103.

---

> > > > ### Comment · Reviewer_18Kq · 2025-08-04
> > > > **Thanks**
> > > >
> > > > I appreciate the additional discussion, which I think is reasonable. I have no further questions.

---

### Official Review · Reviewer_37iR · 2025-07-05

**Clarity:** 2
**Significance:** 2
**Originality:** 2
**Rating:** 3
**Confidence:** 3

**Summary:**

This paper proposes EVDiffuser, a novel diffusion-based framework for consistent low-light video enhancement (LLVE) under varying illumination conditions. It addresses two key limitations of existing methods: (1) sensitivity to illumination fluctuations, and (2) reliance on simultaneous availability of both RGB and event modalities. The core innovations include: 1）A modality-agnostic diffusion pipeline that treats RGB and event data as optional conditions, enabling robust performance even when one modality is missing. 2）Modality-adaptive guidance rescaling to dynamically balance contributions from each sensor based on illumination levels. 3）A new benchmark (V-SDE/V-SDSD) simulating real-world illumination variations. 4）Experiments demonstrate SOTA results on diverse settings (RGB-only, event-only, multimodal).

**Questions:**

1. V-SDE/V-SDSD rely on synthetic illumination curves (sine waves) and simulated noise (Sec. 4). Real-world corner cases (e.g., sudden flashes) may not be fully covered. Please consider to add more experimental results on these real cases.
2. Results on original SDE (see Tab. 6 in main paper) show lower PSNR compared with specialized methods (e.g., EvLight++), suggesting a trade-off between generalization and optimality in fixed conditions. Please explain more on this point.
3. Qualitative results (Fig. 4,8) rely on metrics like LPIPS; human perceptual studies would strengthen claims of "high-quality" reconstruction. Please consider to add more human studies on this point.
4. The auto-regressive inference (Sec. 3.3) stitches video batches using the last predicted frame, risking error propagation in long sequences. Please show more result and analysis on this.

**Ethical Concerns:**

["NO or VERY MINOR ethics concerns only"]

**Final Justification:**

Thanks for the great effort during the rebuttal. The most important issue still not be resolved well that is the too heavy parameters of the proposed model and the too long inference time. So I decided to maintain my rating.

**Limitations:**

1. Runtime profile and parameter count are not provided. These are necessary to justify how compact and efficient the proposed method is.
2. Table 1 only includes image-only baselines up to 2023. Why are more recent methods not included? Moreover, comparisons with video-only methods are also suggested to better evaluate video modeling capacity.
3. Table 2 only evaluates RGB-only baselines on V-SDSD, unlike Table 1 which includes various input modalities (event-only, RGB-only, RGB+event, video+event). This limits the assessment of modality-agnostic performance on diverse datasets.
4. The ablation study is not sufficiently detailed. Have ablations been conducted on LoRA insertion positions, CFG/dropout ratios?
5. Since event streams do not carry color information, generating RGB video from event-only input is inherently ambiguous. Please explain how color priors are learned without RGB supervision.
6. Figure 5 shows that event-only outputs exhibit clear color distortions (e.g., green shifts). Please explain this phenomenon. Suggest evaluating color fidelity using perceptual metrics or conducting user studies.
7. The proposed model is trained only on synthetic event data. Why not include results on real-world event datasets to demonstrate robustness and generalization?

**Paper Formatting Concerns:**

None.

**Quality:**

2

**Strengths And Weaknesses:**

1. The modality-agnostic diffusion framework is the first unified solution for LLVE under illumination variations and partial sensor failure. The design elegantly combines pre-trained video diffusion (CogVideoX) with ControlNet-inspired conditioning.
2. The modality-adaptive guidance rescaling (Eq. 1) is a clever mechanism to exploit sensor-specific strengths (e.g., prioritizing events in low light). Ablation studies (Tab. 3, Fig. 7) validate key components.
3. The V-SDE/V-SDSD datasets address a critical gap in evaluating illumination-varying scenarios, facilitating future research.
4. Zero-shot tests on event-only datasets (Fig. 5) and RGB-only benchmarks (Tab. 2) prove robustness beyond training conditions.

---

> ### Author Rebuttal · Authors · 2025-07-31
>
> We sincerely thank the reviewer for the thorough evaluation and constructive feedback.
>
> **Q1: V-SDE/V-SDSD rely on synthetic illumination curves and may not cover real-world corner cases like sudden flashes.**
>
> We have conducted a detailed analysis of **illumination frequency variations** in our V-SDE dataset (see **Table 7** in the supplementary material), which demonstrates the robustness of our method under extreme illumination changes. In addition to synthetic data, we included real-world, self-captured sequence with varying illumination for qualitative evaluation (**Fig. 8** in the supplementary material), showcasing our method’s ability to generalize to realistic scenarios.
>
> For even more challenging cases - such as sudden flashes, where real-world data are difficult to obtain - we performed additional quantitative experiments on V-SDE-in by **randomly masking frame segments** to simulate abrupt illumination changes. The table below shows that our method maintains robustness under such conditions and significantly outperforms existing approaches.
>
>
> | Method     | PSNR $\uparrow$ | PSNR* $\uparrow$ | SSIM $\uparrow$ | LPIPS $\downarrow$ |
> |------------|------|------|------|-------|
> | EvLight++  | 8.92  | 17.92 | 0.3871 |  0.4102  |
> | EVDiffuser (Ours) | **15.20**  | **23.03** | **0.5492**  | **0.3698** |
>
> **Q2: Performance trade-off between generalization and optimality in fixed conditions.**
>
> We appreciate the question.
> This trade-off is well recognized in the deep learning field (e.g., general-purpose models like GPT versus specialized translation models), where generalization is often prioritized over task-specific optimization due to its practical benefits.
> Our method follows this principle by focusing on generalization across diverse lighting conditions rather than optimizing for fixed low-light scenarios. While specialized models such as EvLight++ may achieve slightly higher PSNR in specific cases, they tend to overfit and lack robustness in broader contexts.
>
> **Q3: Please consider to add more human studies to validate "high-quality" reconstruction claims.**
>
> Thank you for this valuable suggestion. We conducted a user study with 16 participants evaluating the performance of different methods.
> We employ preference ranking to assess user satisfaction across different methods. The evaluation focuses on four key aspects: lighting continuity, video quality, detail preservation, and similarity to the ground truth (GT similarity).
>
> Across all subjects and in every aspect, our method stood out with a consistently superior preference score, highlighting that it is perceived by humans as significantly superior to current baseline methods. We will include these results in the revised manuscript.
>
> | Method | lighting continuity $\downarrow$ | Video Quality $\downarrow$ | Detail Preservation $\downarrow$ | GT similarity $\downarrow$ |
> | --- | --- | --- | --- | --- |
> | ET-Net | 4.375 | 4.9375 | 5 | 4.875 |
> | Retinexformer | 3.8125 | 3.5625 | 3.0625 | 3.6875 |
> | Evlight++ | 3.0625 | 2.4375 | 2.6875 | 2.8125 |
> | EvLowlight | 2.6875 | 3.0625 | 3.125 | 2.5625 |
> | EvLowlight (Ours) | **1** | **1** | **1.125** | **1.0625** |
>
> **Q4: Auto-regressive inference error propagation analysis for long sequences.**
>
> We have demonstrated long-video results (~300 frames) in our supplementary video demo. Technically, unlike existing image-to-video frameworks that rely solely on the first frame and **lack intermediate control**, our approach is constrained by sequential frames and event data. To further prevent over-reliance on the initial frame, we **randomly add noise and mask to the frames** during training (Line 191), encouraging the model to leverage both event streams and RGB information more effectively.
> Overall, these design choices substantially reduce error propagation compared with conventional first-frame-based auto-regressive methods.
>
> **Q5: Runtime profile and parameter count not provided.**
>
> Thank you for the suggestion. The table below summarizes the efficiency profile of our method. Compared to traditional frameworks, our diffusion-based approach requires more parameters but achieves significantly better generalization across diverse scenarios.
>
> | Method        | Params (M)  | Memory (GB) | Inference Time (s/frame) |
> |---------------|------------|-----------|-----------|
> | LLFlow-L-SKF  | 409.50     | 39.91     | 0.30
> | Retinexformer | 15.57      | 1.61      | 0.12
> | ELIE          | 440.32     | 33.36     | 0.32
> | eSL-Net       | 560.94     | 0.56      | 0.18
> | EvLight++     | 225.91     | 26.21     | 0.21
> | EVDiffuser (Ours)   | 7932.80    | 30.89     | 5.63
>
> **Q6: Limited baselines and missing recent methods in Table 1.**
>
> We have expanded our comparison to include more recent methods: MambaLLIE (NeurIPS 2024 [1]) for image-based enhancement, BRVE (CVPR 2024 [2] )and Zero-TIG (arxiv 2025 [3]) for video-based enhancement. Since these methods being optimized for specific low-light scenarios, our method **maintains a significant performance advantage** across all comparisons, as shown below. This expansion will be included in the revision.
>
> | Method                   | PSNR $\uparrow$ | PSNR* $\uparrow$ | SSIM $\uparrow$ | LPIPS $\downarrow$  |
> |--------------------------|-------|-------|--------|--------|
> | MambaLLIE (NeurIPS 2024) | 13.12 | 19.38 | 0.7167 | 0.2973 |
> | BRVE (CVPR 2024)         | 13.96  | 20.40 | 0.6359 | 0.3910 |
> | Zero-TIG (arxiv 2025)    | 9.92  | 12.16 | 0.2218 | 0.7775 |
> | EVDiffuser               | **21.55** | **27.14** | **0.8338** | **0.2249** |
>
> [1] Weng J, Yan Z, et al. Mamballie: Implicit retinex-aware low light enhancement with global-then-local state space[J]. NeurIPS, 2024, 37: 27440-27462.
>
> [2] Zhang G, Zhang Y, et al. Binarized low-light raw video enhancement[C]. CVPR. 2024: 25753-25762.
>
> [3] Li Y, Anantrasirichai N. Zero-TIG: Temporal Consistency-Aware Zero-Shot Illumination-Guided Low-light Video Enhancement[J]. arXiv preprint, 2025.
>
> **Q7: Inconsistent evaluation across modalities in Table 2.**
>
> There may be a misunderstanding regarding Table 2's setting. This table specifically evaluates the **RGB-only** performance on the V-SDSD dataset under varying illumination conditions, demonstrating our framework's generalization capability in single-modality settings. This design choice allows us to isolate and validate the robustness of our modality-agnostic approache.
>
> **Q8: Insufficient ablation study details.**
>
> Sorry for the confusion.
> The ablation study of LoRA on the image branch is presented in Table 4 and Lines 314–318. Regarding the LoRA insertion positions, since we incorporate LoRA only in the image branch, we ablate it by comparing the performance with and without LoRA.
> The CFG ablation results are shown in Figure 7 and Lines 319–342. For the dropout ratio, we followed the commonly used setting in CogVideoX and set it to 0.2. Due to time constraints, we will include more detailed sensitive analysis on the dropout ratios in the revised version.
>
> **Q9: Color prior learning from event-only input without RGB supervision.**
>
> We appreciate this question.
> First, our model is built on a pre-trained video diffusion model (CogVideoX) trained on large-scale RGB datasets, which provides **rich color priors**.
> Secondly, during fine-tuning, we introduce a modality-agnostic training pipeline by applying random RGB masking, forcing the model to **learn semantic content from event data** and learn to associate it with the appropriate color information. This combination of diffusion priors and the novel training strategy enables the model to generate plausible colors even from event-only inputs.
>
> **Q10: Color distortions in event-only outputs (Figure 5).**
>
> We respectfully disagree with the assessment regarding color distortions. Our method is able to recover plausible colors despite their absence in the event data, demonstrating the model’s ability to capture semantic information. In contrast, previous methods always lose color information entirely.
> For quantitative evaluation, event-to-video reconstruction is typically assessed in grayscale, **focusing on structural fidelity rather than color accuracy**. The table below shows quantitative results on the ECD dataset. Our method delivers performance comparable to specialized event-to-video approaches while providing superior generalization across varying illumination conditions and modality availability.
>
> | Method      | MSE $\downarrow$   | SSIM $\uparrow$  | LPIPS $\downarrow$ |
> |-------------|-------|-------|-------|
> | E2VID+      | 0.070 | 0.503 | 0.236 |
> | HyperE2VID  | 0.033 | 0.576 | 0.212 |
> | EVDiffuser (Ours)  | 0.051 | 0.529 | 0.220 |
>
> **Q11: Training only on synthetic event data without real-world datasets.**
>
> There may be a misunderstanding regarding our training datasets. Our model is trained on a combination of synthetic event data and real-world RGB datasets (Line 175-185). Specifically:
>
> * **Event-RGB paired:** The SDE dataset contains **real-world** RGB-event paired data captured using DVIS346C cameras.
> * **RGB-only:** The SDSD dataset is a **real-world** RGB dataset.
> * **Event-only:** The V2E2V dataset is **Synthetic** event data. Employing synthetic event data for event-to-video reconstruction is standard practice, as event cameras capture high-speed motion and extreme lighting conditions that are difficult for RGB cameras to record simultaneously, making paired real-world data collection extremely challenging.
>
> Based on these training datasets, we conduct extensive **quantitative evaluations** on our synthetic datasets, as collecting ground-truth data in real-world illumination-varying scenarios is highly challenging. In addition, we include real-world self-captured sequence for **qualitative evaluation** in the supplementary material (Fig. 8), which demonstrates our method’s ability to generalize to realistic scenarios.

---

> > ### Comment · Reviewer_37iR · 2025-08-07
> > **Thanks for Rebuttal**
> >
> > Thanks for the great effort during the rebuttal. The most important issue still not be resolved well that is the too heavy parameters of the proposed model and the too long inference time. So I decided to maintain my rating.

---

> > > ### Author Response · Authors · 2025-08-07
> > >
> > > We sincerely thank the reviewer for the further discussion.
> > >
> > > We acknowledge the concerns regarding computational efficiency and would like to reiterate the core motivation of our framework. **The modality-agnostic diffusion framework is the first unified solution for LLVE under illumination variations and partial sensor failure**, a highly valuable and practical topic, as noted by all reviewers.
> > > While our unified design entails a higher parameter count and computational cost compared to modality-specific models, it delivers **significantly improved generalization** across a wide range of challenging scenarios. We prioritize generalization and performance over efficiency, and our method focuses on **offline use** as a post-processing technique, rather than for real-time applications.
> > >
> > > Moreover, our proposed framework is inherently **extensible and compatible** with emerging acceleration techniques for diffusion-based generative models. Below, we highlight several promising directions:
> > >
> > > * **Quantized Attention:** Recent advancements [1] in quantized attention mechanisms have demonstrated a 1.8× speedup for CogVideoX without degradation in video quality. These techniques are fully compatible with our CogVideoX-based architecture.
> > >
> > > * **Flow Matching:** The recent flow matching algorithm [2] in video generation reinterprets the denoising process as a multi-stage pyramidal flow, achieving real-time video generation at 24 FPS for 768p resolution. This approach can be seamlessly adapted to our diffusion model with a modality-agnostic training pipeline.
> > >
> > > * **Auto-regressive Modeling:** Recent studies [3,4] have explored auto-regressive strategies for video generation, formulating the task as next-token prediction and achieving real-time sampling at 24 FPS. Extending these techniques to event camera presents a promising direction for fully leveraging the high temporal resolution inherent to event data.
> > >
> > > * **One-step Diffusion:** Recent efforts [5,6,7] have shown progress in reducing the number of denoising steps for generation tasks (achieving a 12 FPS), although they remain limited in handling complex video-generation scenarios. In our preliminary exploration, we implemented a naive few-step diffusion pipeline within our framework, reducing the sampling process to 20 DDIM steps. However, this reduction introduces a trade-off between computational efficiency and video quality, which we recognize as a valuable direction for future work.
> > >
> > > In summary, while our unified framework incurs additional computational cost, it opens up a meaningful research topic and provides a flexible and generalizable solution across modalities and illumination conditions. Furthermore, the aforementioned acceleration techniques are immediately applicable to our proposed EvDiffuser framework and constitute a central focus of our future work. We hope our response helps to clarify the core contributions of our work and resolves the concern.
> > >
> > >
> > > [1] Zhang J, Huang H, Zhang P, et al. Sageattention2: Efficient attention with thorough outlier smoothing and per-thread int4 quantization[J]. arXiv preprint arXiv:2411.10958, 2024.
> > >
> > > [2] Jin Y, Sun Z, Li N, et al. Pyramidal flow matching for efficient video generative modeling[J]. arXiv preprint arXiv:2410.05954, 2024.
> > >
> > > [3] Chen B, Martí Monsó D, Du Y, et al. Diffusion forcing: Next-token prediction meets full-sequence diffusion[J]. Advances in Neural Information Processing Systems, 2024, 37: 24081-24125.
> > >
> > > [4] Lin S, Yang C, He H, et al. Autoregressive Adversarial Post-Training for Real-Time Interactive Video Generation[J]. arXiv preprint arXiv:2506.09350, 2025.
> > >
> > > [5] Ding Z, Jin C, Liu D, et al. Dollar: Few-step video generation via distillation and latent reward optimization[J]. arXiv preprint arXiv:2412.15689, 2024.
> > >
> > > [6] Yin T, Gharbi M, Zhang R, et al. One-step diffusion with distribution matching distillation[C]//Proceedings of the IEEE/CVF conference on computer vision and pattern recognition. 2024: 6613-6623.
> > >
> > > [7] Sauer A, Lorenz D, Blattmann A, et al. Adversarial diffusion distillation[C]//European Conference on Computer Vision. Cham: Springer Nature Switzerland, 2024: 87-103.

---

> ### Author Response · Authors · 2025-08-06
> **Looking Forward to Seeing Your Response!**
>
> Dear reviewer 37iR,
>
> Thanks again for your valuable suggestions! Given the discussion phase is quickly passing, we want to know if our response resolves your concerns. If you have any further questions, we are more than happy to discuss them. We are looking forward to seeing your response!
>
> Best, All authors.

---

### Decision · Program_Chairs · 2025-09-17

**Decision:**

Accept (poster)

**Comment:**

The paper proposed EVDiffuser, a diffusion-based framework for consistent low-light video enhancement (LLVE) under varying illumination conditions. It addresses two key limitations of existing methods: (1) sensitivity to illumination fluctuations, and (2) reliance on simultaneous availability of both RGB and event modalities. The core innovations include: 1）A modality-agnostic diffusion pipeline that treats RGB and event data as optional conditions, enabling robust performance even when one modality is missing. 2）Modality-adaptive guidance rescaling to dynamically balance contributions from each sensor based on illumination levels. 3）A new benchmark (V-SDE/V-SDSD) simulating real-world illumination variations. 4）Experiments demonstrate SOTA results on diverse settings (RGB-only, event-only, multimodal).

The paper was reviewed by four reviewers with diverse comments. The authors submitted rebuttal and there had been discussions among the authors and reviewers. The authors addressed most of the concerns. Afterward, the final ratings were 3XBorderline Accept and 1XBorderline Reject. The only negative reviewer's remaining concerns lie in the heavy parameters of the proposed model and the too long inference time (10 out of 11 comments have been addressed). As provided by the authors, the parameters of the proposed model is 20X other method, which may create concerns for the applications. While all methods have their application scenarios, given the offline nature of the method, it is fine.

I would like to recommendate to accept the paper and the authors are requested to revise the paper accordingly.